# Truncated Variance-Reduced Value Iteration

**Yujia Jin**
Stanford University
yujiajin@stanford.edu

**Ishani Karmarkar**
Stanford University
ishanik@stanford.edu

**Aaron Sidford**
Stanford University
sdiford@stanford.edu

**Jiayi Wang**
Stanford University
jyw@stanford.edu

## Abstract

We provide faster randomized algorithms for computing an $\varepsilon$-optimal policy in a discounted Markov decision process with $\mathcal{A}_{\text{tot}}$-state-action pairs, bounded rewards, and discount factor $\gamma$. We provide an $\tilde{O}(\mathcal{A}_{\text{tot}}[(1-\gamma)^{-3}\varepsilon^{-2} + (1-\gamma)^{-2}])$-time algorithm in the sampling setting, where the probability transition matrix is unknown but accessible through a generative model which can be queried in $\tilde{O}(1)$-time, and an $\tilde{O}(s + \mathcal{A}_{\text{tot}}(1-\gamma)^{-2})$-time algorithm in the offline setting where the probability transition matrix is known and $s$-sparse. These results improve upon the prior state-of-the-art which either ran in $\tilde{O}(\mathcal{A}_{\text{tot}}[(1-\gamma)^{-3}\varepsilon^{-2} + (1-\gamma)^{-3}])$ time ([1, 2]) in the sampling setting, $\tilde{O}(s + \mathcal{A}_{\text{tot}}(1-\gamma)^{-3})$ time ([3]) in the offline setting, or time at least quadratic in the number of states using interior point methods for linear programming. We achieve our results by building upon prior stochastic variance-reduce value iteration methods [1, 2]. We provide a variant that carefully truncates the progress of its iterates to improve the variance of new variance-reduced sampling procedures that we introduce to implement the steps. Our method is essentially model-free and can be implemented in $\tilde{O}(\mathcal{A}_{\text{tot}})$-space when given generative model access. Consequently, our results take a step in closing the sample-complexity gap between model-free and model-based methods.

## 1 Introduction

Markov decision processes (MDPs) are a fundamental mathematical model for decision making under uncertainty. They play a central role in reinforcement learning and prominent problems in computational learning theory (see e.g., [4, 5, 6, 7]). MDPs have been studied extensively for decades ([8, 9]), and there have been numerous algorithmic advances in efficiently optimizing them ([3, 1, 2, 10, 11, 12, 13, 14]).

In this paper, we consider the standard problem of *optimizing a discounted Markov Decision Process (DMDP)* $\mathcal{M} = (\mathcal{S}, \mathcal{A}, \boldsymbol{P}, \boldsymbol{r}, \gamma)$. We consider the *tabular setting* where there is a known finite set of *states* $\mathcal{S}$ and at each state $s \in \mathcal{S}$ there is a finite, non-empty, set of *actions*, $\mathcal{A}_s$ for an agent to choose from; $\mathcal{A} = \{(s, a) : s \in \mathcal{S}, a \in \mathcal{A}_s\}$ denotes the full set of state action pairs and $\mathcal{A}_{\text{tot}} := |\mathcal{A}| \geq |\mathcal{S}|$. The agent proceeds in rounds $t = 0, 1, 2, \ldots$. In each round $t$, the agent is in state $s_t \in \mathcal{S}$; chooses action $a_t \in \mathcal{A}_{s_t}$, which yields a known reward $\boldsymbol{r}_t = \boldsymbol{r}_{s_t,a} \in [0, 1]$; and transitions to random state $s_{t+1}$ sampled (independently) from a (potentially) unknown distribution $\boldsymbol{p}_a(s_t) \in \Delta^{\mathcal{S}}$ for round $t + 1$, where $\boldsymbol{p}_a(s_t)^{\top}$ is the $(s_t, a)$-th row of $\boldsymbol{P} \in [0, 1]^{\mathcal{A} \times \mathcal{S}}$. The goal is to compute an *$\varepsilon$-optimal policy*, where a (deterministic) policy $\pi$, is a mapping from each state $s \in \mathcal{S}$ to an action $\pi(s) \in \mathcal{A}_s$ and is *$\varepsilon$-optimal* if for every initial $s_0 \in \mathcal{S}$ the *expected discounted reward of $\pi$* $\mathbb{E}[\sum_{t \geq 0} r_t \gamma^t]$ is at

least $\boldsymbol{v}_{s_0}^* - \varepsilon$. Here, $\boldsymbol{v}_{s_0}^*$ is the maximum expected discounted reward of any policy applied starting from initial state $s_0$ and $\boldsymbol{v}^* \in \mathbb{R}^{\mathcal{S}}$ is called the *optimal value* of the MDP.

Excitingly, a line of work [15, 16, 2, 3, 17, 18] recently resolved the query complexity for solving DMDPs (up to polylogarithmic factors) in what we call the *sample setting* where the transitions $\boldsymbol{p}_a(s)$ are accessible only through a *generative model* ([16]). A *generative model* is an oracle which when queried with any $s \in \mathcal{S}$ and $a \in \mathcal{A}_s$ returns a random $s' \in \mathcal{S}$ sampled independently from $\boldsymbol{p}_a(s)$ [19]. It was shown in [18] that for all $\varepsilon \in (0, (1-\gamma)^{-1}]$ there is an algorithm which computes an $\varepsilon$-optimal policy with probability $1-\delta$ using $\tilde{O}(\mathcal{A}_{\text{tot}}(1-\gamma)^{-3}\varepsilon^{-2})$ queries where we use $\tilde{O}(\cdot)$ to hide polylogarithmic factors in $\mathcal{A}_{\text{tot}}, \varepsilon^{-1}, (1-\gamma)^{-1}$, and $\delta^{-1}$. This result improved upon a prior result of [17] which achieved the same query complexity for $\varepsilon \in [0, (1-\gamma)^{-1/2}]$, of [2] which achieved this query complexity for $\varepsilon \in [0, 1]$, and of [16] which achieved it for $\varepsilon \in [0, (|\mathcal{S}|(1-\gamma))^{-1/2}]$. This query complexity is known to be optimal in the worst case (up to polylogarithmic factors) due to lower bounds of [16] (and extensions of [20]), which established that the optimal query complexity for finding $\varepsilon$-optimal policies with probability $1-\delta$ is $\Omega(\mathcal{A}_{\text{tot}}(1-\gamma)^{-3}\varepsilon^{-2}\log(\mathcal{A}_{\text{tot}}\delta^{-1}))$.

Interestingly, recent state-of-the-art results [17, 18] (as well as [16]) are *model-based*: they query the oracle for every state-action pair, use the resulting samples to build an empirical model of the MDP, and then solve this empirical model. State-of-the-art computational complexities for the methods are then achieved by applying high-accuracy, algorithms for optimizing MDPs in what we call the *offline setting*, when the transition probabilities are known [2, 17].

Correspondingly, obtaining optimal query complexities for large $\varepsilon$, e.g., $\varepsilon \gg 1$, comes with certain costs. This setting is of interest when the goal is to efficiently compute a coarse approximation of the optimal policy. Model-based methods use space $\Omega(\mathcal{A}_{\text{tot}} \cdot \min((1-\gamma)^{-3}\varepsilon^{-2}, |\mathcal{S}|))$–rather than the $\tilde{O}(\mathcal{A}_{\text{tot}})$ memory used by *model-free* methods (e.g., [2, 3, 21]), which run stochastic, low memory analogs of classic popular algorithms for solving DMDPs (e.g., value iteration). Moreover, although state-of-the-art model-based methods use $\Omega(\mathcal{A}_{\text{tot}}(1-\gamma)^{-3}\varepsilon^{-2})$ *samples,* the state-of-the-art *runtime* to compute the optimal policy is either $\tilde{O}(\mathcal{A}_{\text{tot}}(1-\gamma)^{-3}\max\{1, \varepsilon^{-2}\})$ (using [2]) or has a larger larger polynomial dependence on $\mathcal{A}_{\text{tot}}$ and $|\mathcal{S}|$ by using interior point methods (IPMs) for linear programming (see Section 1.1). Consequently, in the worst case, the *runtime cost* per sample is more than polylogarithmic for $\varepsilon$ sufficiently larger than 1, and it is natural to ask if this can be improved.

These costs are connected to the state-of-the-art runtimes for optimizing DMDPs in the offline setting. Ignoring IPMs (discussed in Section 1.1), the state-of-the-art runtime for optimizing a DMDP is $\tilde{O}(\text{nnz}(\boldsymbol{P}) + \mathcal{A}_{\text{tot}}(1-\gamma)^{-3})$ due to [2] where $\text{nnz}(\boldsymbol{P})$ denotes the number of non-zero entries in $\boldsymbol{P}$, i.e., the number of triplets $(s, s', a)$ where taking action $a \in \mathcal{A}_s$ at state $s \in \mathcal{S}$ has a non-zero probability of transitioning to $s' \in \mathcal{S}$. This method is essentially model-free; it simply performs a variant of stochastic value iteration where passes on $\boldsymbol{P}$ are used to reduce the variance of sampling and can be implemented in $\tilde{O}(\mathcal{A})$-space given access to a generative model and the ability to multiply $\boldsymbol{P}$ with vectors. The difficulty in further improving the runtimes in the sample setting and improving the performance of model-free methods seems connected to the difficulty in improving the additive $\mathcal{A}_{\text{tot}}(1-\gamma)^{-3}$-term in this runtime (see the discussion in Section 1.2.)

In this paper, we ask whether these complexities can be improved. *Is it possible to lower the memory requirements of near-optimal query algorithms for large $\varepsilon$? Can we improve the runtime for optimizing MDPs in the offline setting and can we improve the computational cost per sample in computing optimal policies in DMDPs?* More broadly, *is it possible to close the sample-complexity gap between model-free and model-based methods for optimizing DMDPs?*

## 1.1 Our results

In this paper, we show how to answer each of these motivating questions in the affirmative. We provide faster algorithms for optimizing DMDPs in both the sample and offline setting that are implementable in $\tilde{O}(\mathcal{A}_{\text{tot}})$-space provided suitable access to the input. In addition to computing $\varepsilon$-optimal policies, these methods also compute $\varepsilon$-*optimal values*: we call any $\boldsymbol{v} \in \mathbb{R}^{\mathcal{S}}$ a *value vector* and say that it is $\varepsilon$-optimal if $\|\boldsymbol{v} - \boldsymbol{v}^*\|_\infty \le \varepsilon$.

Here we present our main results on algorithms for solving DMDPs in sample setting and in the offline setting and compare to prior work. For simplicity of comparison, we defer any discussion and comparison of DMDP algorithms that use general IPMs for linear program to the end of this section. The state-of-the-art such IPM methods obtain improved running times but use $\Omega(|\mathcal{S}|^2)$ space and

$\Omega(|\mathcal{S}|^2)$ time and use general-purpose linear system solvers. As such they are perhaps qualitatively different from the more combinatorial or dyanmic-programming based methods, e.g., value iteration and stochastic value iteration, more commonly discussed in this introduction.

In the sample setting, our main result is an algorithm that uses $\tilde{O}(\mathcal{A}_{\text{tot}}[(1-\gamma)^{-3}\varepsilon^{-2} + (1-\gamma)^{-2}])$ samples and time and $O(\mathcal{A}_{\text{tot}})$-space. It improves upon the prior, non-IPM, state-of-the-art which uses $\tilde{O}(\mathcal{A}_{\text{tot}}[(1-\gamma)^{-3}\varepsilon^{-2} + (1-\gamma)^{-3}])$ time [3] and nearly matches the state-of-the-art sample complexity for all $\varepsilon = O((1-\gamma)^{-1/2})$. See Table 2 for a more complete comparison.

**Theorem 1.1.** *In the sample setting, there is an algorithm that uses $\tilde{O}(\mathcal{A}_{\text{tot}}[(1-\gamma)^{-3}\varepsilon^{-2} + (1-\gamma)^{-2}])$ samples and time and $O(\mathcal{A}_{\text{tot}})$ space, and computes an $\varepsilon$-optimal policy and $\varepsilon$-optimal values with probability $1-\delta$.*

Particularly excitingly, the algorithm in Theorem 1.1 runs in time nearly-linear in the number of samples whenever $\varepsilon = O((1-\gamma)^{-1/2})$ and therefore, provided querying the oracle costs $\Omega(1)$, has a near-optimal runtime for such $\varepsilon$! Prior to this work such a near-optimal, non-IPM, runtime (for non-trivially small $\gamma$) was only known for $\varepsilon = \tilde{O}(1)$ ([2]). Similarly, Theorem 1.1 shows that there are model-free algorithms (which for our purposes we define as an $\tilde{O}(\mathcal{A}_{\text{tot}})$ space algorithm) which are nearly-sample optimal whenever $\varepsilon = O((1-\gamma)^{-1/2})$. Previously this was only known for $\varepsilon = \tilde{O}(1)$. As discussed in prior-work ([18, 17]), this large $\varepsilon$ regime is potentially of particular importance in large-scale learning settings, where one would like to *quickly* compute a *coarse* approximation of the optimal policy.

In the offline setting, our main result is an algorithm that uses $\tilde{O}(\text{nnz}(\boldsymbol{P}) + \mathcal{A}_{\text{tot}}(1-\gamma)^{-2})$ time. It improves upon the prior, non-IPM, state-of-the-art which use $\tilde{O}(\text{nnz}(\boldsymbol{P}) + \mathcal{A}_{\text{tot}}(1-\gamma)^{-3})$ time ([2]). See Table 1 for a more complete comparison with prior work.

**Theorem 1.2.** *In the offline setting, there is an algorithm that uses $\tilde{O}(\text{nnz}(\boldsymbol{P}) + \mathcal{A}_{\text{tot}}(1-\gamma)^{-2})$ time, and computes an $\varepsilon$-optimal policy and $\varepsilon$-optimal values with probability $1-\delta$.*

The method of Theorem 1.2 runs in nearly-linear time when $(1-\gamma)^{-1} \leq (\text{nnz}(\boldsymbol{P})/\mathcal{A}_{\text{tot}})^{1/2}$, i.e., the discount factor is not too small relative to the average sparsity of rows of the transition matrix. Prior to this paper, such nearly-linear, non-IPM, runtimes (for non-trivially small $\gamma$) were only known for $(1-\gamma)^{-1} \leq (\text{nnz}(\boldsymbol{P})/\mathcal{A}_{\text{tot}})^{1/3}$ ([2]). Thus, Theorem 1.2 expands the set of DMDPs which can be solved in nearly-linear time. The space usage and input access for this offline algorithm differs from the algorithm in Theorem 1.1 in that the algorithm in Theorem 1.2 assumes that access to the transition $\boldsymbol{P}$ is provided as input and uses this to compute matrix-vector products with value vectors. The algorithm in Theorem 1.2 also requires access to samples from the generative model; if access to the generative model is not provided as input, then using the access to $\boldsymbol{P}$, the algorithm can build a $\tilde{O}(\text{nnz}(\boldsymbol{P}))$ data-structure so that queries to the generative model can be implemented in $\tilde{O}(1)$ time (e.g., see discussion in [2]). Hence, if matrix-vector products and queries to the generative model can be implemented in $\tilde{O}(\mathcal{A}_{\text{tot}})$-space then so can the algorithm in Theorem 1.2.

Table 1: Running times to compute $\varepsilon$-optimal policies in the offline setting. In this table, $E$ denotes an upper bound on the ergodicity of the MDP.

| Algorithm | Runtime | Space |
|:---:|:---:|:---:|
| Value Iteration [22, 11] | $\tilde{O}\left(\text{nnz}(\boldsymbol{P})(1-\gamma)^{-1}\right)$ | $\tilde{O}(\text{nnz}(\boldsymbol{P}))$ |
| Empirical QVI [16] | $\tilde{O}\left(\text{nnz}(\boldsymbol{P}) + \mathcal{A}_{\text{tot}}(1-\gamma)^{-3}\varepsilon^{-2}\right)$ | $\tilde{O}(\text{nnz}(\boldsymbol{P}))$ |
| Randomized Primal-Dual Method [23] | $\tilde{O}\left(\text{nnz}(\boldsymbol{P}) + E\mathcal{A}_{\text{tot}}(1-\gamma)^{-4}\varepsilon^{-2}\right)$ | $\tilde{O}(\mathcal{A}_{\text{tot}})$ |
| High Precision Variance-Reduced Value Iteration [2] | $\tilde{O}\left(\text{nnz}(\boldsymbol{P}) + \mathcal{A}_{\text{tot}}(1-\gamma)^{-3}\right)$ | $\tilde{O}(\mathcal{A}_{\text{tot}})$ |
| Algorithm 4 **This Paper** | $\tilde{O}\left(\text{nnz}(\boldsymbol{P}) + \mathcal{A}_{\text{tot}}(1-\gamma)^{-2}\right)$ | $\tilde{O}(\mathcal{A}_{\text{tot}})$ |

**Exact DMDP Algorithms.** In our our comparison of offline DMDP algorithms in Table 1, we ignored $\mathsf{poly}(\log(\varepsilon^{-1}))$-factors. Consequently, we did not distinguish between algorithms which solve DMDPs to high accuracy, i.e., only depend on $\varepsilon$ polylogarithmically, and those which solve it *exactly*, e.g., have no dependence on $\varepsilon$. There is a line of work on designing such exact methods and the current state-of-the-art is policy iteration, which can be implemented in $\tilde{O}(|\mathcal{S}|^2 \mathcal{A}_{\mathrm{tot}}^2 (1-\gamma)^{-1})$ time ([13, 14]) and a combinatorial interior point method that can be implemented in $\tilde{O}(\mathcal{A}_{\mathrm{tot}}^4)$ time ([10] with no dependence on $\varepsilon$. Note that these methods obtain improved runtime dependence on $\varepsilon$ at the cost of larger dependencies on $|\mathcal{S}|$ and $\mathcal{A}_{\mathrm{tot}}$.

Table 2: Query complexities to compute $\varepsilon$-optimal policy in the sample setting. $M_{\mathrm{erg}}$ denotes an upper bound on the MDP's ergodicity. Here, *model-free* refers to $\tilde{O}(\mathcal{A}_{\mathrm{tot}})$ space methods.

| Algorithm | Queries | $\varepsilon$ range | Model-Free |
|:---:|:---:|:---:|:---:|
| Phased Q-learning [15] | $\tilde{O}\left(\frac{\mathcal{A}_{\mathrm{tot}}}{(1-\gamma)^7 \varepsilon^2}\right)$ | $(0, (1-\gamma)^{-1}]$ | Yes |
| Empirical QVI [16] | $\tilde{O}\left(\frac{\mathcal{A}_{\mathrm{tot}}}{(1-\gamma)^3 \varepsilon^2}\right)$ | $(0, ((1-\gamma)|\mathcal{S}|)^{-1/2}]$ | No |
| Sublinear Variance-Reduced Value Iteration [2] | $\tilde{O}\left(\frac{\mathcal{A}_{\mathrm{tot}}}{(1-\gamma)^4 \varepsilon^2}\right)$ | $(0, (1-\gamma)^{-1/2}]$ | Yes |
| Sublinear Variance-Reduced Q Value Iteration [3] | $\tilde{O}\left(\frac{\mathcal{A}_{\mathrm{tot}}}{(1-\gamma)^3 \varepsilon^2}\right)$ | $(0, 1]$ | Yes |
| Randomized Primal-Dual Method [23] | $\tilde{O}\left(\frac{M_{\mathrm{erg}}\mathcal{A}_{\mathrm{tot}}}{(1-\gamma)^4 \varepsilon^2}\right)$ | $(0, (1-\gamma)^{-1}$ | Yes |
| Empirical MDP + Planning [17] | $\tilde{O}\left(\frac{\mathcal{A}_{\mathrm{tot}}}{(1-\gamma)^3 \varepsilon^2}\right)$ | $(0, (1-\gamma)^{-1/2}]$ | No |
| Perturbed Empirical MDP, Conservative Planning [18] | $\tilde{O}\left(\frac{\mathcal{A}_{\mathrm{tot}}}{(1-\gamma)^3 \varepsilon^2}\right)$ | $(0, (1-\gamma)^{-1}]$ | No |
| Algorithm 5 **This Paper** | $\tilde{O}\left(\frac{\mathcal{A}_{\mathrm{tot}}}{(1-\gamma)^3 \varepsilon^2}\right)$ | $(0, (1-\gamma)^{-1/2}]$ | Yes |

**Comparison with IPM Approaches.** In the offline setting, [2] showed how to reduce solving DMDPs to an $\ell_1$-regression problem in $\boldsymbol{P} \in \mathbb{R}^{\mathcal{A} \times \mathcal{S}}$. For $\ell_1$ regression in a matrix $\boldsymbol{A} \in \mathbb{R}^{n \times d}$ for $n > d$, [12] provides an algorithm that runs in $\tilde{O}(d^{0.5}(\mathrm{nnz}(A) + d^2))$-time, [24] provides an algorithm that runs in $\tilde{O}(nd + d^{2.5})$, and [25, 26, 27] yields an algorithm that runs in $\tilde{O}(\mathcal{A}_{\mathrm{tot}}^\omega)$ time for the current value of the fast matrix multiplication exponent $\omega < 2.371552$ [28]. These offline IPM approaches can be coupled with model-based approaches to yield algorithms in the sample setting. [18] shows that given a DMDP $\mathcal{M}$, with $\tilde{O}\left(\mathcal{A}_{\mathrm{tot}}(1-\gamma)^{-2}\varepsilon^{-3}\right)$ queries to the generative model and time, one can construct a DMDP $\hat{\mathcal{M}}$ such that an optimal policy in $\hat{\mathcal{M}}$ is an $\varepsilon$-optimal for $\mathcal{M}$. Consequently, provided polynomial accuracy in computing the policy suffices, applying the IPMs to $\hat{\mathcal{M}}$ yields runtimes of $\tilde{O}(\mathrm{nnz}(\boldsymbol{P})\sqrt{|\mathcal{S}|} + |\mathcal{S}|^{2.5})$ ([12]), $\tilde{O}(\mathcal{A}_{\mathrm{tot}}|\mathcal{S}| + |\mathcal{S}|^{2.5})$ ([24]), and $\tilde{O}(\mathcal{A}_{\mathrm{tot}}^\omega)$ time [25]. This combination of model-based and IPM-based approaches use super-quadratic time and space, but they may yield better runtimes than Theorem 1.2 in certain regimes where $\gamma$ is sufficiently large relative to $\mathcal{S}$ and $\mathcal{A}_{\mathrm{tot}}$ in the offline setting, or when, additionally, $\varepsilon$ is sufficiently small relative to $\mathcal{S}$ and $\mathcal{A}_{\mathrm{tot}}$ in the sample setting.

## 1.2 Overview of approach

Here we provide an overview of our approach to proving Theorem 1.1 and Theorem 1.2. We motivate our approach from previous methods and discuss the main obstacles and insights needed to obtain our results. For simplicity, we focus on the problem of computing $\varepsilon$-optimal values and discuss computing $\varepsilon$-optimal policies at the end of this section.

**Value iteration.**    Our approach stems from classic value-iteration method ([22, 11]) for computing $\varepsilon$-optimal and its more modern $Q$-value and stochastic counterparts ([16, 3, 29, 30, 31, 32]). As the name suggests, value iteration proceeds in iterations $t = 0, 1, \ldots$ computing *values*, $\boldsymbol{v}^{(t)} \in \mathbb{R}^{\mathcal{S}}$. Starting from initial $\boldsymbol{v}^{(0)} \in \mathbb{R}^{\mathcal{S}}$, in iteration $t \geq 1$, the value vector $\boldsymbol{v}^{(t)}$ is computed as the result of applying the (Bellman) value operator $\mathcal{T} : \mathbb{R}^{\mathcal{S}} \mapsto \mathbb{R}^{\mathcal{S}}$, i.e.,

$$\boldsymbol{v}^{(t)} \leftarrow \mathcal{T}(\boldsymbol{v}^{(t-1)}) \text{ where } \mathcal{T}(\boldsymbol{v})(s) := \max_{a \in \mathcal{A}_s}(\boldsymbol{r}_a(s) + \gamma \boldsymbol{p}_a(s)^{\top} \boldsymbol{v}) \text{ for all } s \in \mathcal{S} \text{ and } \boldsymbol{v} \in \mathbb{R}^{\mathcal{S}}. \quad (1)$$

It is well-known that the value operator is $\gamma$-contractive and therefore, $\|\mathcal{T}(\boldsymbol{v}) - \boldsymbol{v}^*\|_{\infty} \leq \gamma \|\boldsymbol{v} - \boldsymbol{v}^*\|_{\infty}$ for all $v \in \mathbb{R}^{\mathcal{S}}$ ([11, 22, 2]). If we initialize $\boldsymbol{v}^{(0)} = \boldsymbol{0}$ then since $\|\boldsymbol{v}^*\|_{\infty} \leq (1-\gamma)^{-1}$ [22, 11], we see that $\|\boldsymbol{v}^{(t)} - \boldsymbol{v}^*\|_{\infty} \leq \gamma^t \|\boldsymbol{v}^{(0)} - \boldsymbol{v}^*\|_{\infty} \leq \gamma^t (1-\gamma)^{-1} \leq (1-\gamma)^{-1} \exp(-t(1-\gamma))$. Thus, $\boldsymbol{v}^{(t)}$ are $\varepsilon$-optimal values for any $t \geq (1-\gamma)^{-1} \log(\varepsilon^{-1}(1-\gamma)^{-1})$. This yields an $\tilde{O}(\mathrm{nnz}(\boldsymbol{P})(1-\gamma)^{-1})$ time algorithm in the offline setting.

**Stochastic value iteration and variance reduction.**    To improve on the runtime of value iteration and apply it in the sample setting, a line of work implements *stochastic* variants of value iteration ([16, 2, 3, 23, 17, 18]). Those methods take approximate value iteration steps where the *expected utilities* $\boldsymbol{p}_a(s)^{\top} \boldsymbol{v}$ in (1) for each state-action pair are replaced by a *stochastic estimate* of the expected utilities. In particular, note that $\boldsymbol{p}_a(s)^{\top} \boldsymbol{v} = \mathbb{E}_{i \sim \boldsymbol{p}_a(s)} \boldsymbol{v}_i$, i.e., the expected value of $\boldsymbol{v}_i$ where $i$ is drawn from the distribution given by $\boldsymbol{p}_a(s)$. This is compatible in the sample setting, as computing $\boldsymbol{v}_i$ for $i$ drawn from $\boldsymbol{p}_a(s)$ yields an unbiased estimate of $\boldsymbol{p}_a(s)^{\top} \boldsymbol{v}$ with 1 query and $O(1)$ time.

State-of-the-art model-free methods in the sample setting ([3]) and non-IPM runtimes in the offline setting ([3]) improve further by more carefully approximating the *expected utilities* $\boldsymbol{p}_a(s)^{\top} \boldsymbol{v}$ of each state-action pair $(s, a) \in \mathcal{A}$. Broadly, given an arbitrary $\boldsymbol{v}^{(0)}$ they first compute $\boldsymbol{x} \in \mathbb{R}^{\mathcal{A}}$ that approximates $\boldsymbol{P}\boldsymbol{v}^{(0)}$, i.e., $\boldsymbol{x}_a(s)$ approximates $[\boldsymbol{P}\boldsymbol{v}^{(0)}]_{(s,a)} = \boldsymbol{p}_a(s)^{\top} \boldsymbol{v}^{(0)}$ for all $(s, a) \in \mathcal{A}$. In the offline setting, $\boldsymbol{x} = \boldsymbol{P}\boldsymbol{v}^{(0)}$ can be computed directly in $O(\mathrm{nnz}(\boldsymbol{P}))$-time. In the sample setting, $\boldsymbol{x} \approx \boldsymbol{P}\boldsymbol{v}^{(0)}$ can be approximated to sufficient accuracy using multiple queries for each state-action pair. Then, in each iteration $t \geq 1$ of the algorithm, fresh samples are taken to compute $\boldsymbol{g}^{(t)} \approx \boldsymbol{P}(\boldsymbol{v}^{(t-1)} - \boldsymbol{v}^{(0)})$ and perform the following update:

$$\boldsymbol{v}^{(t)}(s) \leftarrow \max_{a \in \mathcal{A}_s}(\boldsymbol{r}_a(s) + \gamma(\boldsymbol{x}_a(s) + \boldsymbol{g}_a(s)^{(t)})) \text{ for all } s \in \mathcal{S} \text{ and } \boldsymbol{v} \in \mathbb{R}^{\mathcal{S}}. \quad (2)$$

This approach is advantageous because sampling errors for estimating $\boldsymbol{P}(\boldsymbol{v}^{(t-1)} - \boldsymbol{v}^{(0)})$ depend on the magnitude of $\boldsymbol{v}^{(t-1)} - \boldsymbol{v}^{(0)}$. After approximately computing $\boldsymbol{x}$, the remaining task of computing $\boldsymbol{g}^{(t)} \approx \boldsymbol{P}(\boldsymbol{v}^{(t-1)} - \boldsymbol{v}^{(0)})$ so that $\boldsymbol{x} + \boldsymbol{g}^{(t)} \approx \boldsymbol{P}\boldsymbol{v}^{(t-1)}$ may be easier than the task of directly estimating $\boldsymbol{P}\boldsymbol{v}^{(t)}$ (since $\boldsymbol{v}^{(t-1)} - \boldsymbol{v}^{(0)}$ is smaller in magnitude than $\boldsymbol{v}^{(t)}$ entrywise.) Due to similarities of this approach to variance-reduced optimization methods, e.g. ([33, 34]), this technique is called *variance reduction* [2].

The works [2, 3], showed that if $\boldsymbol{x}$ is computed sufficiently accurately and $\boldsymbol{v}^{(0)}$ are $\alpha$-optimal values then applying (2) for $t = \Theta((1-\gamma)^{-1})$ yields $\boldsymbol{v}^{(t)}$ that is $\alpha/2$-optimal in just $\tilde{O}(\mathcal{A}_{\mathrm{tot}}(1-\gamma)^{-3})$ time and samples! [2] leverages this technique to compute $\varepsilon$-optimal values in the offline setting in $\tilde{O}(\mathrm{nnz}(\boldsymbol{P}) + \mathcal{A}_{\mathrm{tot}}(1-\gamma)^{-3})$ time. [3] uses a similar approach to compute $\varepsilon$-optimal values in $\tilde{O}(\mathcal{A}_{\mathrm{tot}}[(1-\gamma)^{-3}\varepsilon^{-2} + (1-\gamma)^{-3})$ time and samples in the sample setting. A key difference in [2] and [3] is the accuracy to which they must approximate the initial utility $\boldsymbol{x} \approx \boldsymbol{P}\boldsymbol{v}^{(0)}$.

**Recursive variance reduction.**    To improve upon the prior model-free approaches of [2, 3] we improve how exactly the variance reduction is performed. We perform a similar scheme as in (2) and use essentially the same techniques as in [3, 2] towards estimating $\boldsymbol{x}$. Where we differ from prior work is in how we estimate the change in approximate utilities $\boldsymbol{g}^{(t)} \approx \boldsymbol{P}(\boldsymbol{v}^{(t-1)} - \boldsymbol{v}^{(0)})$. Rather than *directly* sampling to estimate this difference we instead sample to estimate each individual $\boldsymbol{P}(\boldsymbol{v}^{(t-1)} - \boldsymbol{v}^{(t)})$ and maintain the sum. Concretely, for $t \geq 1$, we compute $\boldsymbol{\Delta}^{(t)}$ such that

$$\boldsymbol{\Delta}^{(t)} \approx \boldsymbol{P}(\boldsymbol{v}^{(t)} - \boldsymbol{v}^{(t-1)}) \quad (3)$$

so that these recursive approximations telescope. More precisely, setting $\boldsymbol{g}^{(0)} = \boldsymbol{0}$, for $t \geq 1$, we set

$$\boldsymbol{g}^{(t)} \leftarrow \boldsymbol{g}^{(t-1)} + \boldsymbol{\Delta}^{(t-1)} \approx \boldsymbol{P}(\boldsymbol{v}^{(t-2)} - \boldsymbol{v}^{(0)}) + \boldsymbol{P}(\boldsymbol{v}^{(t-1)} - \boldsymbol{v}^{(t-2)}) = \boldsymbol{P}(\boldsymbol{v}^{(t-1)} - \boldsymbol{v}^{(0)}). \quad (4)$$

This difference is perhaps similar to how methods such as SARAH ([34]) differ from SVRG ([33]). Consequently, we similarly call this approximation scheme *recursive variance reduction*. Interestingly, in constrast to the finite sum setting considered in [33, 34], in our setting, recursive variance reduction for solving DMDPs ultimately leads to direct quantitative improvements on worst case complexity.

To analyze this recursive variance reduction method, we treat the error in $\boldsymbol{g}^{(t)} \approx \boldsymbol{P}(\boldsymbol{v}^{(t-1)} - \boldsymbol{v}^{(0)})$ as a martingale and analzye it using Freedman's inequality [35] (as stated in [36]). The hope in applying this approach is that by better bounding and reasoning about the changes in $\boldsymbol{v}^{(t)}$, better bounds on the error of the sampling could be obtained by leveraging structural properties of the iterates.

*Unfortunately*, without further information about the change in $\boldsymbol{v}^{(t)}$ or larger change to the analysis of variance reduced value iteration, in the worst case, the variance can be too large for this approach to work naively. Concretely, prior work ([2]) showed that it sufficed to maintain that $\|\boldsymbol{g}^{(t+1)} - \boldsymbol{P}\boldsymbol{v}^{(t)}\|_\infty \leq O((1-\gamma)\alpha)$. However, imagine that $\boldsymbol{v}^* = \alpha\boldsymbol{1}$, $\boldsymbol{v}^{(0)} = \boldsymbol{0}$, and in each iteration $t$ one coordinate of $\boldsymbol{v}^{(t)} - \boldsymbol{v}^{(t-1)}$ is $\Omega(\alpha)$. If $|\mathcal{S}| \approx (1-\gamma)^{-1}$ and $\|\boldsymbol{p}_a(s)\|_\infty = O(1/|\mathcal{S}|)$ for some $(s, a) \in \mathcal{A}$ then the variance of each sample used to estimate $\boldsymbol{p}_a(s)^\top (\boldsymbol{v}^{(t)} - \boldsymbol{v}^{(t-1)}) = \Omega(1/|\mathcal{S}|) = \Omega((1-\gamma))$. Applying Freedman's inequality, e.g., [36], and taking $b$ samples for each $O((1-\gamma)^{-1})$ iteration would yield, roughly, $\|\boldsymbol{g}^{(t+1)} - \boldsymbol{P}(\boldsymbol{v}^{(t)} - \boldsymbol{v}^{(0)})\|_\infty = O((1-\gamma)^{-1}(1-\gamma)/\sqrt{b}) = O(1/\sqrt{b})$. Consequently $b = \Omega((1-\gamma)^{-2})$ and $\Omega((1-\gamma)^{-3})$ samples would be needed in total, i.e., there is no improvement. Next, we will discuss how we circumvent this obstacle by *combining* recursive variance reduction with a *second* algorithm technique, which we call *truncation*.

**Truncated-value iteration.** The key insight to make our new recursive variance reduction scheme for value iteration yield faster runtimes is to modify the value iteration scheme itself. Recall that in the previous paragraph, we described that the case challenging case for recursive variance reduction occurs when, for example, in every iteration, a single coordinate of $v$ changes by $\Omega(\alpha)$. We observe that there is a simple modification that one could make to value iteration to ensure that there is not such a large change between each iteration; simply *truncate* the change in each iteration so that no coordinate of $\boldsymbol{v}^{(t)}$ changes too much! To motivate our algorithm, consider the following *truncated* variant of value iteration where

$$\boldsymbol{v}^{(t)} = \text{median}(\boldsymbol{v}^{(t-1)} - (1-\gamma)\alpha, \mathcal{T}(\boldsymbol{v}^{(t-1)}), \boldsymbol{v}^{(t-1)} + (1-\gamma)\alpha) \tag{5}$$

Where $\text{median}$ applies the median of the arguments entrywise. In other words, suppose we apply value iteration where we decrease or *truncate* the change from $\boldsymbol{v}^{(t-1)}$ to $\boldsymbol{v}^{(t)}$ so that it is no more than $(1 - \gamma)\alpha$ in absolute value in any coordinate. Then, provided that $\boldsymbol{v}^{(t)}$ is $\alpha$-optimal, we can show that it is still the case that $\|\boldsymbol{v}^{(t)} - \boldsymbol{v}^*\|_\infty \leq \gamma\|\boldsymbol{v}^{(t-1)} - \boldsymbol{v}^*\|_\infty$. In other words, the worst-case progress of value iteration is unaffected! This follows immediately from the fact that $\|\boldsymbol{v}^{(t)} - \boldsymbol{v}^*\|_\infty \leq \gamma\|\boldsymbol{v}^{(t-1)} - \boldsymbol{v}^*\|_\infty$ in value iteration and the following simple technical lemma.

**Lemma 1.3.** *For $\boldsymbol{a}, \boldsymbol{b}, \boldsymbol{x} \in \mathbb{R}^n$ and $\gamma, \alpha > 0$, let $\boldsymbol{c} := \text{median}\{\boldsymbol{a} - (1-\gamma)\alpha\boldsymbol{1}, \boldsymbol{b}, \boldsymbol{a} + (1-\gamma)\alpha\boldsymbol{1}\}$, where median is applied entrywise. Then, if $\|\boldsymbol{b} - \boldsymbol{x}\|_\infty \leq \gamma\|\boldsymbol{a} - \boldsymbol{x}\|_\infty$ and $\|\boldsymbol{a} - \boldsymbol{x}\|_\infty \leq \alpha$, then $\|\boldsymbol{c} - \boldsymbol{x}\|_\infty \leq \gamma\|\boldsymbol{a} - \boldsymbol{x}\|_\infty$.*

Applying truncated value iteration, we know that $\|\boldsymbol{v}^{(t)} - \boldsymbol{v}^{(t-1)}\|_\infty \leq (1-\gamma)\alpha$. In other words, the worst-case change in a coordinate has decreased by a factor of $(1 - \gamma)$! We show that this smaller movement bound does indeed decrease the variance in the martingale when using the aforementioned averaging scheme. We show this truncation scheme, when *combined* with our recursive variance reduction scheme (4) for estimating $\boldsymbol{P}(\boldsymbol{v}^{(t)} - \boldsymbol{v}^{(0)})$, reduces the total samples required to estimate this and halve the error from $\tilde{O}((1-\gamma)^{-3})$ to just $\tilde{O}((1-\gamma)^{-2})$ per state-action pair.

**Our method.** Our algorithm applies stochastic truncated value iteration using sampling to estimate each $\boldsymbol{g}^{(t)} \approx \boldsymbol{P}(\boldsymbol{v}^{(t)} - \boldsymbol{v}^{(0)})$ as described. Some minor additional modifications are needed, however, to obtain our results. Perhaps the most substantial is our use of the *monotonicity technique*, as in prior work ([2, 3]). That is, we modify our method so that each $\boldsymbol{v}^{(t)}$ is always an *underestimate* of $\boldsymbol{v}^*$ and the $\boldsymbol{v}^{(t)}$ *increase* monotonically as $t$ increases. Thus, we only truncate the increase in the $\boldsymbol{v}^{(t)}$ (since they do not decrease, and the median operation in (5) reduces to a minimum in Lemma 1.3).

Beyond simplifying this aspect of the algorithm, as in prior work, this monotonicity technique allows us to *simultaneously* compute an $\varepsilon$-approximate policy as well as an $\varepsilon$-optimal value vector. We do this by tracking the actions associated with changed $\boldsymbol{v}^{(t)}$ values, i.e., the $\text{argmax}$ in (2) in a variable

| **Algorithm 1:** Sample$(\boldsymbol{u}, \boldsymbol{p}, M, \eta)$ | **Algorithm 2:** ApxUtility$(\boldsymbol{u}, M, \eta)$ |
|---|---|
| **Input:** Value vector $\boldsymbol{u} \in \mathbb{R}^{\mathcal{S}}, \boldsymbol{p} \in \Delta^{\mathcal{S}}$, sample size $M$, and offset parameter $\eta \geq 0$. | **Input:** Value vector $\boldsymbol{u} \in \mathbb{R}^{S}$, sample size $M$, and offset parameter $\eta \geq 0$. |
| 1 **for** *each* $n \in [M]$ **do** | 1 **for** *each* $(s, a) \in \mathcal{A}$ **do** |
| 2    Choose $i_n \in \mathcal{S}$ independently with $\mathbb{P}\{i_n = t\} = \boldsymbol{p}(t)$; |    // In the sample setting, $\boldsymbol{p}_a(s)$ is passed implicitly. |
| 3 $x = \frac{1}{M} \sum_{n \in [M]} \boldsymbol{u}(i_n)$; | 2    $\boldsymbol{x}_a(s) = $ Sample$(\boldsymbol{u}, \boldsymbol{p}_a(s), M, \eta)$; |
| 4 $\hat{\sigma} = \frac{1}{M} \sum_{n \in [M]} (\boldsymbol{u}(i_n))^2 - x^2$; | |
| 5 $\tilde{x} \leftarrow x - \sqrt{2\eta\hat{\sigma}} - 4\eta^{3/4} \|\boldsymbol{u}\|_\infty - (2/3)\eta \|\boldsymbol{u}\|_\infty$; | 3 **return** $\boldsymbol{x}$ |
| 6 **return** $\tilde{x}$ | |

$\pi^{(t)}$, which denotes the current estimated policy in iteration $t$ of value iteration. Concretely, the monotonicity technique allows us to maintain the invariant that at each iteration $t$, the current value estimate and policy estimate $\pi^{(t)}, \boldsymbol{v}^{(t)}$ satisfy the relation $\boldsymbol{v}^{(t)} \leq \mathcal{T}[\boldsymbol{v}^{(t)}]$. Note that this ensures that the value of $\pi^{(t)}$ (denoted $\boldsymbol{v}^{\pi^{(t)}}$) is *at least* $\boldsymbol{v}^{(t)}$ because

$$\boldsymbol{v}^{(t)} \leq \mathcal{T}[\boldsymbol{v}^{(t)}] \leq \mathcal{T}^2[\boldsymbol{v}^{(t)}] \leq \cdots \mathcal{T}^\infty[\boldsymbol{v}^{(t)}] = \boldsymbol{v}^{\pi^{(t)}}$$

Thus, whenever $\boldsymbol{v}^{(t)}$ is an $\varepsilon$-optimal value, $\pi^{(t)}$ is an *at least* $\varepsilon$-optimal policy.

By computing initial expected utilities $\boldsymbol{x} = \boldsymbol{P}\boldsymbol{v}^{(0)}$ exactly, we obtain our offline results. By carefully estimating $\boldsymbol{x} \approx \boldsymbol{P}\boldsymbol{v}^{(0)}$ as in [3] we obtain our sampling results. Finally, building off of the analysis of [37] for deterministic or highly-mixing MDPs, we also show our method obtains even faster convergence guarantees under additional non-worst-case assumptions on the MDP structure.

### 1.3 Notation and paper outline

**General notation.** Caligraphic upper case letters denote sets and operators, lowercase boldface letters denote vectors, and uppercase boldface letters (e.g., $\boldsymbol{P}, \boldsymbol{I}$) denote matrices. $\boldsymbol{0}$ and $\boldsymbol{1}$ denote the all-ones and all-zeros vectors, $[m] := \{1, ...., m\}$, and $\Delta^n := \{x \in \mathbb{R}^n : \boldsymbol{0} \leq x$ and $\|x\|_1 = 1\}$ is the simplex. For $\boldsymbol{v} \in \mathbb{R}^{\mathcal{S}}$, we use $\boldsymbol{v}_i$ or $\boldsymbol{v}(i)$ for the $i$-th entry of vector $\boldsymbol{v}$. For vectors $\boldsymbol{v} \in \mathbb{R}^{\mathcal{A}}$, we use $\boldsymbol{v}_a(s)$ to denote the $(s, a)$-th entry of $\boldsymbol{v}$, where $(s, a) \in \mathcal{A}$. We use $\sqrt{\boldsymbol{v}}, \boldsymbol{v}^2, |\boldsymbol{v}| \in \mathbb{R}^n$ for the element-wise square root, square, and absolute value of $\boldsymbol{v}$ respectively and $\max\{\boldsymbol{u}, \boldsymbol{v}\}$ and $\text{median}\{\boldsymbol{u}, \boldsymbol{v}, \boldsymbol{w}\}$ for element-wise maximum and median respectively. For $\boldsymbol{v}, \boldsymbol{x} \in \mathbb{R}^n$, $\boldsymbol{v} \leq \boldsymbol{x}$ denotes that $\boldsymbol{v}(i) \leq \boldsymbol{x}(i)$ for each $i \in [n]$ (analogously for $<, \geq, >$.) We call $\boldsymbol{x} \in \mathbb{R}^n$ an $\alpha$-*underestimate* of $\boldsymbol{y} \in \mathbb{R}^n$ if $\boldsymbol{y} - \alpha\boldsymbol{1} \leq \boldsymbol{x} \leq \boldsymbol{y}$ for $\alpha \geq 0$ (see the discussion of monotonicity in Section 1.2 for motivation).

**DMDP.** As discussed, the objective in optimizing a DMDP is to find an $\varepsilon$-approximate policy $\pi$ and values. For a policy $\pi$, we use $\mathcal{T}_\pi(\boldsymbol{u}) : \mathbb{R}^{\mathcal{S}} \mapsto \mathbb{R}^{\mathcal{S}}$ to denote the value operator associated with $\pi$, i.e., $\mathcal{T}_\pi(\boldsymbol{u})(s) := \boldsymbol{r}_{\pi(s)}(s) + \gamma\boldsymbol{p}_{\pi(s)}(s)^\top \boldsymbol{u}$ for all value vectors $\boldsymbol{u} \in \mathbb{R}^{\mathcal{S}}$ and $s \in \mathcal{S}$. We let $\boldsymbol{v}^\pi$ denote the unique value vector such that $\mathcal{T}_\pi(\boldsymbol{v}^\pi) = \boldsymbol{v}^\pi$ and define its variance as $\boldsymbol{\sigma}_{\boldsymbol{u}^\pi} := \boldsymbol{P}^\pi(\boldsymbol{u}^\pi)^2 - (\boldsymbol{P}^\pi\boldsymbol{u}^\pi)^2$, where $\boldsymbol{P}^\pi \in \mathbb{R}^{\mathcal{S}\times\mathcal{S}}$ is the matrix such that $\boldsymbol{P}^\pi_{s,s'} = \boldsymbol{P}_{s,\pi(s)}(s')$. The *optimal value vector* $\boldsymbol{v}^\star \in \mathbb{R}^{\mathcal{S}}$ of the optimal policy $\pi^\star$ is the unique vector with $\mathcal{T}(\boldsymbol{v}^\star) = \boldsymbol{v}^\star$, and $\boldsymbol{P}^\star \in \mathbb{R}^{\mathcal{S}\times\mathcal{S}} := \boldsymbol{P}^{\pi^\star}$.

**Outline.** Section 2 presents our offline setting results and Section 3 our sample setting results. Section A discusses specialized settings where we can obtain even faster convergence guarantees. Omitted proofs are deferred to Appendix B.

## 2 Offline algorithm

In this section, we present our high-precision algorithm for finding an approximately optimal policy in the offline setting. We first define Sample (Algorithm 1), which approximately computes products between $\boldsymbol{p} \in \Delta^S$ and a value vector $\boldsymbol{u} \in \mathbb{R}^{\mathcal{S}}$ using samples from a generative model. The following lemma states some immediate estimation bounds on Sample using linearity and the fact that $\boldsymbol{p} \in \Delta^{\mathcal{S}}$.

**Lemma 2.1.** *Let* $x = $ Sample$(\boldsymbol{u}, \boldsymbol{p}, M, 0)$ *for* $\boldsymbol{p} \in \Delta^n$, $M \in \mathbb{Z}_{>0}$, $\varepsilon > 0$, *and* $\boldsymbol{u} \in \mathbb{R}^{\mathcal{S}}$. *Then,* $\mathbb{E}[x] = \boldsymbol{p}^\top\boldsymbol{u}$, $|x| \leq \|\boldsymbol{u}\|_\infty$, *and* $\text{Var}[x] \leq 1/M \|\boldsymbol{u}\|_\infty^2$.

We can naturally apply `Sample` to each state-action pair in $\mathcal{M}$ as in the subroutine `ApxUtility` (Algorithm 2). If $\boldsymbol{x} = \texttt{ApxUtility}(\boldsymbol{u}, M, \eta)$, then $\boldsymbol{x}(s,a)$ is an estimate of the expected utility of taking action $a \in \mathcal{A}_s$ from state $s \in \mathcal{S}$ (as discussed in Section 1.2). When $\eta > 0$, this estimate may potentially be shifted to increase the probability that $\boldsymbol{x}$ underestimates the true changes in utilities; we leverage this in Section 3 (see also the discussion of monotonicity in Section 1.2). The terms arising in the definition of $\tilde{x}$ arise from applying Bernstein's inequality (Theorem B.2) to guarantee that $\tilde{x} \le x - \eta$ with high probability.

The following algorithm `TVRVI` (Algorithm 3) takes as input an initial value vector $\boldsymbol{v}^{(0)}$ and policy $\pi^{(0)}$ such that $\boldsymbol{v}^{(0)}$ is an $\alpha$-underestimate of $\boldsymbol{v}^\star$ along with an approximate offset vector $\boldsymbol{x}$, which is a $\beta$-underestimate of $\boldsymbol{P}\boldsymbol{v}^{(0)}$. It runs runs $L = \tilde{O}((1-\gamma)^{-1})$ iterations of approximate value iteration, making one call to `Sample`(Algorithm 1) with a sample size of $M = \tilde{O}((1-\gamma)^{-1})$ in each iteration. The algorithm outputs $\boldsymbol{v}^L$ which we show is an $\alpha/2$-underestimate of $\boldsymbol{v}^\star$ (Corollary 2.5).

`TVRVI` (Algorithm 3) is similar to variance reduced value iteration [2], in that each iteration, we draw $M$ samples and use `Sample` to maintain underestimates of $\boldsymbol{p}_a(s)^\top(\boldsymbol{v}^{(\ell)} - \boldsymbol{v}^{(\ell-1)})$ for each sate-action pair $(s,a)$. However, there are two key distinctions between `TVRVI`and variance-reduced value iteration [2] that enable our improvement. First, we use the recursive variance reduction technique, as described by (3) and (4), and second we apply truncation (Line 7), which essentially implements the truncation described in Lemma 1.3. Lemma 2.2 below illustrates how these two techniques can be combined to bound the necessary sample complexity for maintaining approximate transitions $\boldsymbol{p}_a(s)^\top(\boldsymbol{w}^{(t)} - \boldsymbol{w}^{(0)})$ for a general sequence of $\ell_\infty$-bounded vectors $\{\boldsymbol{w}^{(i)}\}_{i=1}^T$. The analysis leverages Freedman's Inequality [35] as stated in [36] and restated in Theorem B.1.

**Lemma 2.2.** *Let $T \in \mathbb{Z}_{>0}$ and $\boldsymbol{w}^{(0)}, \boldsymbol{w}^{(1)}, ..., \boldsymbol{w}^{(T)} \in \mathbb{R}^\mathcal{S}$ such that $\left\|\boldsymbol{w}^{(i)} - \boldsymbol{w}^{(i-1)}\right\|_\infty \le \tau$ for all $i \in [T]$. Then, for any $\boldsymbol{p} \in \Delta^\mathcal{S}$, $\delta \in (0,1)$, and $M \ge 2^8 T \log(2/\delta)$ with probability $1 - \delta$, $\left|\boldsymbol{p}^\top(\boldsymbol{w}^{(t)} - \boldsymbol{w}^{(0)}) - \sum_{i\in[t]} \sum_{j\in[M]} \texttt{Sample}(\boldsymbol{w}^{(i)} - \boldsymbol{w}^{(i-1)}, \boldsymbol{p}, 1, 0) \cdot 1/M\right| \le \tau/8$ for all $t \in [T]$.*

---

**Algorithm 3:** $\texttt{TVRVI}(\boldsymbol{v}^{(0)}, \pi^{(0)}, \boldsymbol{x}, \alpha, \delta)$

**Input:** Initial values $\boldsymbol{v}^{(0)} \in \mathbb{R}^S$, which is an $\alpha$-underestimate of $\boldsymbol{v}^\star$.
**Input:** Initial policy $\pi^{(0)}$ such that $\boldsymbol{v}^{(0)} \le \mathcal{T}_{\pi^{(0)}}(\boldsymbol{v}^{(0)})$.
**Input:** Accuracy $\alpha \in [0, (1-\gamma)^{-1}]$ and failure probability $\delta \in (0,1)$.
**Input:** Offsets $\boldsymbol{x} \in \mathbb{R}^\mathcal{A}$ ;                   // entrywise underestimate of $\boldsymbol{P}\boldsymbol{v}^{(0)}$

1 Initialize $\boldsymbol{g}^{(1)} \in \mathbb{R}^\mathcal{A}$ and $\hat{\boldsymbol{g}}^{(1)} \in \mathbb{R}^\mathcal{A}$ to $\boldsymbol{0}$;
2 $L = \lceil \log(8)(1-\gamma)^{-1} \rceil$ and $M = \lceil L \cdot 2^8 \log(2\mathcal{A}_{\text{tot}}/\delta) \rceil$ ;
3 **for** *each iteration $\ell \in [L]$* **do**
4 $\quad$ $\tilde{\boldsymbol{Q}} = \boldsymbol{r} + \gamma(\boldsymbol{x} + \hat{\boldsymbol{g}}^{(\ell)})$;
5 $\quad$ $\boldsymbol{v}^{(\ell)} = \boldsymbol{v}^{(\ell-1)}$ and $\pi^{(\ell)} = \pi^{(\ell-1)}$ ;
6 $\quad$ **for** *each state $i \in \mathcal{S}$* **do**
$\quad\quad$ // Compute truncated value update (and associated action)
7 $\quad\quad$ $\tilde{\boldsymbol{v}}^{(\ell)}(i) = \min\{\max_{a\in\mathcal{A}_i} \tilde{\boldsymbol{Q}}_{i,a}, \boldsymbol{v}^{(\ell-1)} + (1-\gamma)\alpha\}$ and $\tilde{\pi}_i^{(\ell)} = \text{argmax}_{a\in\mathcal{A}_i} \tilde{\boldsymbol{Q}}_{i,a}$;
$\quad\quad$ // Update value and policy if it improves
8 $\quad\quad$ **if** $\tilde{\boldsymbol{v}}^{(\ell)}(i) \ge \boldsymbol{v}^{(\ell)}(i)$ **then** $\boldsymbol{v}^{(\ell)}(i) = \tilde{\boldsymbol{v}}^{(\ell)}(i)$ and $\pi_i^{(\ell)} = \tilde{\pi}_i^{(\ell)}$ ;
$\quad$ // Update for maintaining estimates of $\boldsymbol{P}(\boldsymbol{v}^{(l)} - \boldsymbol{v}^0)$.
9 $\quad$ $\boldsymbol{\Delta}^{(\ell)} = \texttt{ApxUtility}(\boldsymbol{v}^{(\ell)} - \boldsymbol{v}^{(\ell-1)}, M, 0)$ and $\boldsymbol{g}^{(\ell+1)} = \boldsymbol{g}^{(\ell)} + \boldsymbol{\Delta}^{(\ell)}$ ;
$\quad$ // Shift estimates so that $\hat{\boldsymbol{g}}^{(\ell+1)}$ always *underestimates* $\boldsymbol{p}_a(s)^\top \boldsymbol{v}^{(\ell)}$ .
10 $\quad$ $\hat{\boldsymbol{g}}^{(\ell+1)} = \boldsymbol{g}^{(\ell+1)} - \frac{(1-\gamma)\alpha}{8}\boldsymbol{1}$;
11 **return** $(\boldsymbol{v}^{(L)}, \pi^{(L)})$

---

While it is unclear how to significantly improve the constant of $2^8 = 256$ appearing in Lemma 2.2 (and consequently Algorithm 3), we note that tightening these constants in the application of Freedman's inequality could be of practical interest. By applying Lemma 2.2 to the iterates $\boldsymbol{v}^{(\ell)}$ in `TVRVI`, the following Corollary 2.3 shows that we can maintain additive $O((1-\gamma)\alpha)$-underestimates of the

transitions $\boldsymbol{p}_a(s)^\top(\boldsymbol{v}^{(\ell)} - \boldsymbol{v}^{(0)})$ using only $\tilde{O}(L)$ samples (as opposed to the $\tilde{O}(L^2)$ samples required in [2]) per state-action pair.

**Corollary 2.3.** *In* `TVRVI` *(Algorithm 3), with probability* $1 - \delta$*, in Lines 9, 10 and 2, for all* $s \in \mathcal{S}, a \in \mathcal{A}_s$*, and* $\ell \in [L]$*, we have* $\left| \boldsymbol{g}_a^{(\ell)}(s) - \boldsymbol{p}_a(s)^\top(\boldsymbol{v}^{(\ell-1)} - \boldsymbol{v}^{(0)}) \right| \leq (1 - \gamma)\alpha/8$ *and therefore* $\hat{\boldsymbol{g}}_a^{(\ell)}$ *is a* $(1 - \gamma)\alpha/4$*-underestimate of* $\boldsymbol{p}_a(s)^\top(\boldsymbol{v}^{(\ell-1)} - \boldsymbol{v}^{(0)})$.

The following Lemma 2.4 shows that whenever the event in Corollary 2.3 holds, `TVRVI` (Algorithm 3) is approximately contractive and maintains monotonicity of the approximate values. By accumulating the error bounds in Lemma 2.4, we also obtain the following Corollary 2.5, which guarantees that `TVRVI` halves the error in the initial estimate $\boldsymbol{v}^{(0)}$.

**Lemma 2.4.** *Suppose that for some* $\boldsymbol{\beta} \in \mathbb{R}_{\geq 0}^{\mathcal{A}}$*,* $\boldsymbol{P}\boldsymbol{v}^{(0)} - \boldsymbol{\beta} \leq \boldsymbol{x} \leq \boldsymbol{P}\boldsymbol{v}^{(0)}$ *and let* $\boldsymbol{\beta}_{\pi^\star} \in \mathbb{R}^{\mathcal{S}}$ *be defined as* $\boldsymbol{\beta}_{\pi^\star}(s) := \boldsymbol{\beta}_{\pi^\star(s)}(s)$ *for each* $s \in \mathcal{S}$*. Then, with probability* $1 - \delta$*, at the end of every iteration* $\ell \in [L]$ *(Line 3) in* `TVRVI`$(\boldsymbol{v}^{(0)}, \pi^{(0)}, \boldsymbol{x}, \alpha, \delta)$*, the following hold for* $\boldsymbol{\xi} := \gamma((1 - \gamma)\alpha/4\mathbf{1} + \boldsymbol{\beta}_{\pi^\star})$:

$$\boldsymbol{v}^{(\ell-1)} \leq \boldsymbol{v}^{(\ell)} \leq \mathcal{T}_{\pi^{(\ell)}}(\boldsymbol{v}^{(\ell)}), \tag{6}$$

$$0 \leq \boldsymbol{v}^\star - \boldsymbol{v}^{(\ell)} \leq \max\left(\gamma\boldsymbol{P}^\star(\boldsymbol{v}^\star - \boldsymbol{v}^{(\ell-1)}) + \boldsymbol{\xi}, \gamma(\boldsymbol{v}^\star - \boldsymbol{v}^{(\ell-1)})\right). \tag{7}$$

**Corollary 2.5.** *Suppose that for some* $\alpha \geq 0$ *and* $\boldsymbol{\beta} \in \mathbb{R}_{\geq 0}^{\mathcal{A}}$*,* $\boldsymbol{P}\boldsymbol{v}^{(0)} - \boldsymbol{\beta} \leq \boldsymbol{x} \leq \boldsymbol{P}\boldsymbol{v}^{(0)}$*;* $\boldsymbol{v}^{(0)}$ *is an* $\alpha$*-underestimate of* $\boldsymbol{v}^\star$*; and* $\boldsymbol{v}^{(0)} \leq \mathcal{T}_{\pi^{(0)}}(\boldsymbol{v}^{(0)})$*. Let* $\boldsymbol{\beta}_{\pi^\star} \in \mathbb{R}^{\mathcal{S}}$ *be defined as* $\boldsymbol{\beta}_{\pi^\star}(s) := \boldsymbol{\beta}_{\pi^\star(s)}(s)$ *for each* $s \in \mathcal{S}$*. Let* $(\boldsymbol{v}^{(L)}, \pi^{(L)}) = $ `TVRVI`$(\boldsymbol{v}^{(0)}, \pi^{(0)}, \alpha, \delta)$*, and* $L, M$ *be as in Line 2. Define* $\boldsymbol{\xi} := \gamma\left((1 - \gamma)\alpha/4 \cdot \mathbf{1} + \boldsymbol{\beta}_{\pi^\star}\right)$*. Then, with probability* $1 - \delta$*,* $\mathbf{0} \leq \boldsymbol{v}^\star - \boldsymbol{v}^{(L)} \leq \gamma^L\alpha \cdot \mathbf{1} + (\boldsymbol{I} - \gamma\boldsymbol{P}^\star)^{-1}\boldsymbol{\xi}$*, and* $\boldsymbol{v}^{(L)} \leq \mathcal{T}_{\pi^{(L)}}(\boldsymbol{v}^{(L)})$*. In particular, if* $\boldsymbol{\beta} = \mathbf{0}$*, then for* $L > \log(8)(1 - \gamma)^{-1}$ *we can reduce the error in* $\boldsymbol{v}^{(0)}$ *by half:* $\mathbf{0} \leq \boldsymbol{v}^\star - \boldsymbol{v}^{(L)} \leq (\boldsymbol{v}^\star - \boldsymbol{v}^{(0)})/2$*. Additionally,* `TVRVI` *is implementable with* $\tilde{O}(\mathcal{A}_{\text{tot}}ML)$ *sample queries to the generative model and time and* $O(\mathcal{A}_{\text{tot}})$ *space.*

Theorem 1.2 now follows by recursively applying Corollary 2.5. `OfflineTVRVI` (Algorithm 4) provides the pseudocode for the algorithm guaranteed by Theorem 1.2.

---

**Algorithm 4:** `OfflineTVRVI`$(\varepsilon, \delta)$

---

**Input:** Target precision $\varepsilon$ and failure probability $\delta \in (0, 1)$

1    $K = \lceil \log_2(\varepsilon^{-1}(1 - \gamma)^{-1}) \rceil$, $\boldsymbol{v}_0 = \mathbf{0}$, $\pi_0$ is an arbitrary policy, and $\alpha_0(1 - \gamma)^{-1}$;

2    **for** *each iteration* $k \in [K]$ **do**

3       $\alpha_k = \alpha_{k-1}/2 = 2^{-k}(1 - \gamma)^{-1}$;

4       $\boldsymbol{x} = \boldsymbol{P}\boldsymbol{v}_{k-1}$ ;

5       $(\boldsymbol{v}_k, \pi_k) = $ `TVRVI`$(\boldsymbol{v}_{k-1}, \pi_{k-1}, \boldsymbol{x}, \alpha_{k-1}, 0, \delta/K)$;

6    **return** $(\boldsymbol{v}_K, \pi_K)$

---

## 3   Sample setting algorithm

In this section, we show how to extend the analysis in the previous section in the sample setting, where we do not have explicit access to $\boldsymbol{P}$. We follow a similar framework as in [3] to show that we can instead estimate the offsets $\boldsymbol{x}$ in `OfflineTVRVI` by taking additional samples from the generative model. The pseudocode is shown in `SampleTVRVI`(Algorithm 5.) To analyze the algorithm, we first bound the error incurred when approximating the exact offsets $\boldsymbol{x}$ in Line 4 of `OfflineTVRVI` (Algorithm 4) with approximate offsets $\tilde{\boldsymbol{x}} \approx \boldsymbol{P}\boldsymbol{v}_{k-1}$ computed by sampling from the generative model. The proof leverages Hoeffding's and Bernstein's inequality, and follows a similar structure as the proof of Lemma 5.1 of [3].

**Algorithm 5:** SampleTVRVI($\varepsilon, \delta$)

---

**Input:** Target precision $\varepsilon$ and failure probability $\delta \in (0, 1)$

1   $K = \lceil \log_2(\varepsilon^{-1}(1 - \gamma)^{-1}) \rceil$ ;

2   $v_0 = \mathbf{0}$, $\pi_0$ is an arbitrary policy, and $\alpha_0 = (1 - \gamma)^{-1}$;

3   **for** *each iteration* $k \in [K]$ **do**

4      $\alpha_k = \alpha_{k-1}/2 = 2^{-k}(1 - \gamma)^{-1}$ ;

5      $N = 6500(1 - \gamma)^{-3} \log(8\mathcal{A}_{\text{tot}} K \delta^{-1})$;

6      $N_{k-1} = N \max((1 - \gamma), \alpha_{k-1}^{-2})$ ;

7      $\eta_{k-1} = N_{k-1}^{-1} \log(8\mathcal{A}_{\text{tot}} K \delta^{-1})$ ;

8      $x_k = \text{ApxUtility}(v_{k-1}, N_{k-1}, \eta_{k-1})$;

9      $(v_k, \pi_k) = \text{TVRVI}(v_{k-1}, \pi_{k-1}, x_k, \alpha_{k-1}, \delta/K)$ ;

10  **return** $(v_K, \pi_K)$

---

**Lemma 3.1.** *Consider* $u \in \mathbb{R}^{\mathcal{S}}$. *Let* $x = \text{ApxUtility}(u, m \cdot \mathcal{A}_{\text{tot}}, \eta)$, $m \geq \log(1/2\delta^{-1})$, *and* $\eta = (m\mathcal{A}_{\text{tot}})^{-1} \log(1/2\delta^{-1})$. *Then, with probability* $1 - \delta$,

$$Pu - 2\sqrt{2\eta \boldsymbol{\sigma}_{v^\star}} + \left(2\sqrt{2\eta} \|u - v^\star\|_\infty + 18\eta^{3/4} \|u\|_\infty\right) \leq x \leq Pu.$$

Finally, to obtain our main result Theorem 1.1, we utilize worst-case bounds on $\boldsymbol{\sigma}_{v^\star}$ from prior work [1] (see Lemma B.3, Lemma B.4) and inductively apply Lemma 3.1 and Corollary 2.5.

The constant of 6500 appearing in the initialization of $N$ in Algorithm 5 arises due to technical reasons, from applying Bernstein's inequality, Hoeffding's inequality, union bound over all $K$ outer loop iterations, and bounds on $\boldsymbol{\sigma}_{v^\star}$ from prior work [3] to prove Lemma 3.1. While it is unclear how to directly further tighten this constant, the proof of Lemma 3.1 shows that in the expression $N = 6500(1 - \gamma)^{-3} \log(8\mathcal{A}_{\text{tot}} K \delta^{-1})$ there is a natural trade-off between the leading constant (in this case 6500) and the number of outer loop iterations $K$. By increasing the number of outer-loop iterations $K$ by constants, one can relax the error requirements of each iteration (i.e., decrease $N$ by constants at the cost of increased logarithmic dependence on $|S|$, $\mathcal{A}_{\text{tot}}$). Although not the primary focus of our work, such trade-offs might be of practical importance.

## 4   Conclusion

We provided faster and more space-efficient algorithms for solving DMDPs. We showed how to apply truncation and recursive variance reduction to improve upon prior variance-reduced value iterations methods. Ultimately, these techniques reduced an additive $\tilde{O}((1 - \gamma)^{-3})$ term in the time and sample complexity of prior variance-reduced value iteration methods to $\tilde{O}((1 - \gamma)^{-2})$.

Natural open problems left by our work include exploring the practical implications of our techniques and exploring whether further runtime improvements are possible. For example, it may be of practical interest to explore whether there exist other analogs of truncation that do not need to limit the progress in individual steps of value iteration. Additionally, the question of whether the $\tilde{O}((1 - \gamma)^{-2})$ term in our time and sample complexities can be further improved to $\tilde{O}((1 - \gamma)^{-1})$ is a natural open problem; an affirmative answer to this question would yield the first near-optimal running times for solving a DMDP with a generative model for all $\varepsilon$ and fully bridge the sample complexity gap between model-based and model-free methods. We hope this paper supports further studying these questions and establishing the optimal runtime for solving MDPs.

## Acknowledgements

Thank you to Yuxin Chen for interesting and motivating discussion about model-based methods in RL. Thank you to the anonymous reviewers for their helpful feedback. Yujia Jin and Ishani Karmarkar were funded in part by NSF CAREER Award CCF-1844855, NSF Grant CCF-1955039, and a PayPal research award. Aaron Sidford was funded in part by a Microsoft Research Faculty Fellowship, NSF CAREER Award CCF-1844855, NSF Grant CCF1955039, and a PayPal research award. Part of this work was conducted while visiting the Simons Institute for the Theory of Computing. Yujia Jin's contributions to the project occurred while she was a graduate student at Stanford.

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

# A  Faster problem-dependent convergence

In this section, we propose a modified version of the `SampleTVRVI` algorithm, named `ProblemDependentTVRVI`. This algorithm adjusts the number of required samples based on the structure of the MDP under consideration. Inspired by [38], we then consider MDPs with small ranges of optimal values and the extreme case of highly mixing MDPs in which state transitions are sampled from a fixed distribution.

Note that in the proof of Theorem 1.1, the error during convergence caused by approximations of values is bounded by $(\boldsymbol{I}-\gamma\boldsymbol{P}^\star)^{-1}\boldsymbol{\xi}_k$ for $\boldsymbol{\xi}_k \leq \frac{(1-\gamma)\alpha_k}{4}\boldsymbol{1}+2\sqrt{2\eta_k}\boldsymbol{\sigma}_{\boldsymbol{v}^\star}+(2\sqrt{2\eta_k}\alpha_k+18\eta_k^{3/4}\left\|\boldsymbol{v}^{(0)}\right\|_\infty)\boldsymbol{1}$. In its proof, we upper bound the variance term $\left\|(\boldsymbol{I}-\gamma\boldsymbol{P}^\star)^{-1}\sqrt{\boldsymbol{\sigma}_{\boldsymbol{v}^\star}}\right\|_\infty$ by $3(1-\gamma)^{-1.5}$ using Lemma B.4. However, as $\alpha_k$ decreases and the variance term becomes dominant, a number of samples proportional to the size of the variance term suffices to control the error during each iteration. Given $V$ which upper bounds $\left\|(\boldsymbol{I}-\gamma\boldsymbol{P}^\star)^{-1}\sqrt{\boldsymbol{\sigma}_{\boldsymbol{v}^\star}}\right\|_\infty$, we can further refine `SampleTVRVI` to reduce the number of samples taken after an initial burn-in phase and obtain improved complexities when $V$ is signficantly small. Hence, we obtain the following Algorithm 6 and Theorem A.1.

---

**Algorithm 6:** `ProblemDependentTVRVI`$(\varepsilon, \delta, V)$

---

**Input:** Target precision $\varepsilon$, failure probability $\delta \in (0,1)$, and $V \geq \left\|(\boldsymbol{I}-\gamma\boldsymbol{P}^\star)^{-1}\sqrt{\boldsymbol{\sigma}_{\boldsymbol{v}^\star}}\right\|_\infty$.

1 $K = \lceil\log_2(\varepsilon^{-1}(1-\gamma)^{-1})\rceil$ ;

2 $\boldsymbol{v}_0 = \boldsymbol{0}$, $\pi_0$ is an arbitrary policy, and $\alpha_0 = \frac{1}{1-\gamma}$;

3 **for** *each iteration $k \in [K]$* **do**

4     $\alpha_k = \alpha_{k-1}/2 = 2^{-k}(1-\gamma)^{-1}$ ;

5     **if** $k < \lceil\log_2\left(\frac{128(1-\gamma)^{-5}}{V^3}\right)\rceil$ **then**

6        $N_{k-1} = 6500 \cdot (1-\gamma)^{-3}\max((1-\gamma),\alpha_{k-1}^{-2})\log(8\mathcal{A}_{\text{tot}}K\delta^{-1})$ ;       // Burn-in phase

7     **else**

8        $N_{k-1} = 1024 \cdot \alpha_{k-1}^{-2}V^2\log(8\mathcal{A}_{\text{tot}}K\delta^{-1})$ ;       // Variance-dependent phase

9     $\eta_{k-1} = N_{k-1}^{-1}\log(8\mathcal{A}_{\text{tot}}K\delta^{-1})$ ;

10     $\boldsymbol{x}_k = \texttt{ApxUtility}(\boldsymbol{v}_{k-1},N_{k-1},\eta_{k-1})$;

11     $(\boldsymbol{v}_k,\pi_k) = \texttt{TVRVI}(\boldsymbol{v}_{k-1},\pi_{k-1},\boldsymbol{x}_k,\alpha_{k-1},\delta/K)$;

12 **return** $(\boldsymbol{v}_K,\pi_K)$

---

**Theorem A.1.** *In the sample setting, there is an algorithm (Algorithm 6) that, given $3(1-\gamma)^{-1.5} \geq V \geq \left\|(\boldsymbol{I}-\gamma\boldsymbol{P}^\star)^{-1}\sqrt{\boldsymbol{\sigma}_{\boldsymbol{v}^\star}}\right\|_\infty$, uses $\tilde{O}\left(\mathcal{A}_{\text{tot}}\left(\varepsilon^{-2}V^2+(1-\gamma)^{-2}\right)\right)$ samples and time and $O(\mathcal{A}_{\text{tot}})$ space, and computes an $\varepsilon$-optimal policy and $\varepsilon$-optimal values with probability $1 - \delta$.*

*Proof.* Let $K$, $\alpha_k$, and $(\boldsymbol{v}_k,\pi_k)$ be as defined in Lines 1, 4, and 11 of `ProblemDependentTVRVI`$(\varepsilon, \delta, V)$.

For the correctness of the algorithm, we first induct on $k$ to show that for each $k \in [K]$, with probability $1 - k\delta/K$,

$$\boldsymbol{0} \leq \boldsymbol{v}^\star - \boldsymbol{v}^{\pi_k} \leq \boldsymbol{v}^\star - \boldsymbol{v}_k \leq \alpha_k, \quad \text{and } \boldsymbol{v}_k \leq \mathcal{T}_{\pi_k}(\boldsymbol{v}_k).$$

The base case is trivial, as $\boldsymbol{0} \leq \boldsymbol{v}^\star - \boldsymbol{v}^{\pi_0} \leq \boldsymbol{v}^\star - \boldsymbol{v}_0 \leq (1-\gamma)^{-1}\boldsymbol{1}$.

For the inductive step, observe that by Lemma 3.1, we see that with probability $1 - \delta/K$,

$$\boldsymbol{P}\boldsymbol{v}_{k-1} - \left[2\sqrt{2\eta_{k-1}\boldsymbol{\sigma}_{\boldsymbol{v}^\star}} + \left(2\sqrt{2\eta_{k-1}}\alpha_{k-1} + 18\eta_{k-1}^{3/4}\left\|\boldsymbol{v}_{k-1}\right\|_\infty\right)\boldsymbol{1}\right] \leq \boldsymbol{x}_k \leq \boldsymbol{P}\boldsymbol{v}_{k-1}. \quad (8)$$

Additionally, by the inductive hypothesis, with probability $1 - (k-1)\delta/K$,

$$0 \leq \boldsymbol{v}^\star - \boldsymbol{v}^{\pi_{k-1}} \leq \boldsymbol{v}^\star - \boldsymbol{v}_k \leq \gamma^L\alpha_{k-1}\cdot\boldsymbol{1} + (\boldsymbol{I}-\gamma\boldsymbol{P}^\star)^{-1}\boldsymbol{\xi}_{k-1} \leq \alpha_k\boldsymbol{1}, \quad \text{and } \boldsymbol{v}_k \leq \mathcal{T}_{\pi_k}(\boldsymbol{v}_k). \quad (9)$$

Thus, by union bound, with probability $1 - k\delta/K$, both (8) and (9) hold. We condition on this event in the remainder of the inductive step.

Now, we apply Corollary 2.5 with

$$\beta = 2\sqrt{2\eta_{k-1}\boldsymbol{\sigma_{v^\star}}} + \left(2\sqrt{2\eta_{k-1}}\alpha_{k-1} + 18\eta_{k-1}^{3/4}\left\|\boldsymbol{v}_{k-1}\right\|_\infty\right)\mathbf{1}.$$

Consequently, we have

$$0 \le \boldsymbol{v}^\star - \boldsymbol{v}_k \le \gamma^L \alpha_{k-1}\cdot\mathbf{1} + (\boldsymbol{I} - \gamma\boldsymbol{P}^\star)^{-1}\boldsymbol{\xi}_{k-1} \le \frac{\alpha_{k-1}}{8}\mathbf{1} + (\boldsymbol{I} - \gamma\boldsymbol{P}^\star)^{-1}\boldsymbol{\xi}_{k-1},$$

for $\boldsymbol{\xi}_{k-1} \le \frac{(1-\gamma)\alpha_{k-1}}{4}\mathbf{1} + 2\sqrt{2\eta_{k-1}\boldsymbol{\sigma_{v^\star}}} + \left(2\sqrt{2\eta_{k-1}}\alpha_{k-1} + 18\eta_{k-1}^{3/4}\left\|\boldsymbol{v}_{k-1}\right\|_\infty\right)\mathbf{1}$, and $\boldsymbol{v}_k \le \mathcal{T}_{\pi_k}(\boldsymbol{v}_k)$.

Note that $(\boldsymbol{I} - \gamma\boldsymbol{P}^\star)^{-1}\mathbf{1} \le \frac{1}{1-\gamma}\mathbf{1}$. Hence, if $k < \lceil\log_2(1-\gamma)^{-5}/V^3\rceil$, we use Lemma B.4 along with the facts that $(\boldsymbol{I} - \gamma\boldsymbol{P}^\star)^{-1}\mathbf{1} = 1/(1-\gamma)\mathbf{1}$ and the choice of $\eta_{k-1}$ to obtain (identical to the proof of Theorem 1.1):

$$(\mathbf{I} - \gamma\boldsymbol{P}^\star)^{-1}\boldsymbol{\xi}_{k-1} \le \left[\frac{\alpha_{k-1}}{4} + 2\sqrt{6\frac{\eta_{k-1}}{(1-\gamma)^3}} + 2\sqrt{\frac{2(1-\gamma)^3\min((1-\gamma)^{-1},\alpha_{k-1}^2)}{6500(1-\gamma)^2}}\alpha_{k-1}\right]\mathbf{1}$$

$$+ \left[18\left(\frac{((1-\gamma)^3\min((1-\gamma)^{-1},\alpha_{k-1}^2))}{6500(1-\gamma)^{8/3}}\right)^{3/4}\right]\mathbf{1}$$

$$\le [\alpha_{k-1}/4 + 2\sqrt{6/6500}\cdot\alpha_{k-1} + 2\sqrt{2/6500}(1-\gamma)^{1/2}\min((1-\gamma)^{-1/2},\alpha_{k-1})\alpha_{k-1}$$

$$+ 18\cdot(10^{-3})(1-\gamma)^{1/4}\min((1-\gamma)^{-3/4},\alpha_{k-1}^{3/2})]\mathbf{1}$$

$$\le [\alpha_{k-1}/4 + 4\sqrt{6/6500}\cdot\alpha_{k-1} + 18\cdot(10^{-3})\alpha_{k-1}]\mathbf{1} \le \frac{3}{8}\alpha_{k-1}\mathbf{1}.$$

If instead $k \ge \lceil\log_2(1-\gamma)^{-5}/V^3\rceil$, then $\alpha_k \le \frac{1}{128}(1-\gamma)^4 V^3$, and $\eta_{k-1} = \alpha_{k-1}^2/(1024\cdot V^2)$. Consequently,

$$(\boldsymbol{I} - \gamma\boldsymbol{P}^\star)^{-1}\boldsymbol{\xi}_{k-1} \le 2\sqrt{2\eta_{k-1}}(\mathbf{I} - \gamma\boldsymbol{P}^\star)^{-1}\sqrt{\boldsymbol{\sigma_{v^\star}}}$$

$$+ \left[\frac{\alpha_{k-1}}{4} + 2\sqrt{2\eta_{k-1}}(\mathbf{I} - \gamma\boldsymbol{P}^\star)^{-1}\alpha_{k-1} + 18\eta_{k-1}^{3/4}(\mathbf{I} - \gamma\boldsymbol{P}^\star)^{-1}\left\|\boldsymbol{v}_{k-1}\right\|_\infty\right]\mathbf{1}$$

$$\le \frac{\alpha_{k-1}}{4}\mathbf{1} + \frac{2\sqrt{2}\alpha_{k-1}}{4(1-\gamma)\sqrt{1024}V}V\mathbf{1} + \frac{18}{(1-\gamma)^2}\left(\frac{\alpha_{k-1}^2}{1024\cdot V^2}\right)^{3/4}\mathbf{1}$$

$$\le \left[\frac{\alpha_{k-1}}{8} + \frac{\alpha_{k-1}}{4}\right]\mathbf{1} \le \frac{3}{8}\alpha_{k-1}\mathbf{1}.$$

Therefore in either case,

$$\boldsymbol{v}^\star - \boldsymbol{v}_{k-1} \le \frac{\alpha_{k-1}}{2}\mathbf{1} = \alpha_k\mathbf{1}.$$

Moreover, we can use that $\boldsymbol{v}_k \le \mathcal{T}_{\pi_k}(\boldsymbol{v}_k)$ to see that

$$\boldsymbol{v}_k \le \mathcal{T}_{\pi_k}(\boldsymbol{v}_k) \le \mathcal{T}_{\pi_k}^2(\boldsymbol{v}_k) \le \cdots \le \mathcal{T}_{\pi_k}^\infty(\boldsymbol{v}_k) = \boldsymbol{v}^{\pi_k} \le \boldsymbol{v}^\star.$$

This completes the inductive step.

Consequently, taking $k = K = \lceil\log_2(\varepsilon^{-1}(1-\gamma)^{-1})\rceil$ iterations of the outer loop, with probability $1 - \delta$, we have that $0 \le \boldsymbol{v}^\star - \boldsymbol{v}^{\pi_K} \le \boldsymbol{v}^\star - \boldsymbol{v}_K \le \alpha_K \le \varepsilon$ and

$$\boldsymbol{v}_k \le \mathcal{T}_{\pi_k}(\boldsymbol{v}_k) \le \mathcal{T}_{\pi_k}^2(\boldsymbol{v}_k) \le \cdots \le \mathcal{T}_{\pi_k}^\infty(\boldsymbol{v}_k) = \boldsymbol{v}^{\pi_k} \le \boldsymbol{v}^\star,$$

that is, $\boldsymbol{v}_k$ is an $\varepsilon$-optimal value and $\pi_K$ is an $\varepsilon$-optimal policy.

The total number of samples and time required is $\tilde{O}\left(\mathcal{A}_{\text{tot}}\left(\varepsilon^{-2}V^2 + (1-\gamma)^{-2}\right)\right)$. For the space complexity, note that the algorithm can be implemented to maintain only $O(1)$ vectors in $\mathbb{R}^{\mathcal{A}_{\text{tot}}}$. ∎

Theorem A.1 yields improved complexities for solving MDPs when $\left\|(\boldsymbol{I} - \gamma\boldsymbol{P}^\star)^{-1}\sqrt{\boldsymbol{\sigma_{v^\star}}}\right\|_\infty$ is nontrivially bounded. Following [37] we mention two particular such settings where we can apply Theorem A.1 to obtain better problem-dependent sample and runtime bounds than Theorem 1.1.

**Deterministic MDPs** For a deterministic MDP, each action deterministically transitions to a single state. That is, for all $(s, a) \in \mathcal{A}$, $\boldsymbol{p}_a(s) = \mathbf{1}_{s'}$ (the indicator vector of $s' \in \mathcal{S}$) for some $s' \in \mathcal{S}$. In this case, $\boldsymbol{\sigma}_{\boldsymbol{v}^\star} = \mathbf{0}$. Consequently, if the MDP is deterministic, the algorithm converges with just $\tilde{O}((1-\gamma)^3)$ samples to the generative model and time. We note that in this setting of deterministic MDPs, there may be alternative approaches to obtain the same or better runtime and sample complexity.

**Small range.** Define the range of optimal values for a MDP as $\mathrm{rng}(\boldsymbol{v}^*) \stackrel{\text{def}}{=} \max_{s \in \mathcal{S}} \boldsymbol{v}_s^* - \min_{s \in \mathcal{S}} \boldsymbol{v}_s^*$. Note that $\boldsymbol{\sigma}_{\boldsymbol{v}^*} \leq \mathrm{rng}(\boldsymbol{v}^*)^2 \mathbf{1}$. So, $\left\| (\boldsymbol{I} - \gamma \boldsymbol{P}^\star)^{-1} \sqrt{\boldsymbol{\sigma}_{\boldsymbol{v}^\star}} \right\|_\infty \leq (1-\gamma)^{-1} \mathrm{rng}(\boldsymbol{v}^*)$. Therefore, by Theorem A.1, given an approximate upper bound of $\left\| (\boldsymbol{I} - \gamma \boldsymbol{P}^\star)^{-1} \sqrt{\boldsymbol{\sigma}_{\boldsymbol{v}^\star}} \right\|_\infty$ our algorithm is implementable with $\tilde{O}(\mathcal{A}_{\text{tot}}(\varepsilon^{-2}(1-\gamma)^{-2} \mathrm{rng}(\boldsymbol{v}^\star)^2 + (1-\gamma)^{-2}))$ samples and time.

**Highly mixing domains.** [37] showed that a contextual bandit problem can be modeled as an MDP where the next state is sampled from a fixed stationary distribution. Using the fact that the transition function is independent of the prior state and action, the authors of [38] show that $\mathrm{rng}(\boldsymbol{v}^*) \leq 1$ with a simple proof in its Appendix A.2. Hence, by the argument in the preceding paragraph $\tilde{O}\left(\mathcal{A}_{\text{tot}}\left(\varepsilon^{-2}(1-\gamma)^{-2} + (1-\gamma)^{-2}\right)\right)$ samples and time suffice in this setting.

# B Omitted proofs from the main body

## B.1 Omitted proof of Lemma 1.3

**Lemma 1.3.** *For $\boldsymbol{a}, \boldsymbol{b}, \boldsymbol{x} \in \mathbb{R}^n$ and $\gamma, \alpha > 0$, let $\boldsymbol{c} := \mathrm{median}\{\boldsymbol{a} - (1-\gamma)\alpha\mathbf{1}, \boldsymbol{b}, \boldsymbol{a} + (1-\gamma)\alpha\mathbf{1}\}$, where median is applied entrywise. Then, if $\|\boldsymbol{b} - \boldsymbol{x}\|_\infty \leq \gamma \|\boldsymbol{a} - \boldsymbol{x}\|_\infty$ and $\|\boldsymbol{a} - \boldsymbol{x}\|_\infty \leq \alpha$, then $\|\boldsymbol{c} - \boldsymbol{x}\|_\infty \leq \gamma \|\boldsymbol{a} - \boldsymbol{x}\|_\infty$.*

*Proof.* Consider the $i$-th entry $(c-x)_i$. There are three cases.

First, suppose $a_i - (1-\gamma)\alpha \leq b_i \leq a_i + (1-\gamma)\alpha$. Then, $|c_i - x_i| = |b_i - x_i| \leq \gamma \|\boldsymbol{a} - \boldsymbol{x}\|_\infty$

Second, suppose $b_i \leq a_i - (1-\gamma)\alpha \leq a_i + (1-\gamma)\alpha$. Then, $c_i - x_i \geq b_i - x_i \geq - \|\boldsymbol{b} - \boldsymbol{x}\|_\infty \geq -\gamma \|\boldsymbol{a} - \boldsymbol{x}\|_\infty$. Meanwhile, $c_i - x_i = a_i - (1-\gamma)\alpha - x_i \leq \|\boldsymbol{a} - \boldsymbol{x}\|_\infty - (1-\gamma)\alpha$. Now, because $\|\boldsymbol{a} - \boldsymbol{x}\|_\infty \leq \alpha$, we have that $(1-\gamma)\|\boldsymbol{a} - \boldsymbol{x}\|_\infty \leq (1-\gamma)\alpha$. So, $\|\boldsymbol{a} - \boldsymbol{x}\|_\infty - (1-\gamma)\alpha \leq \gamma \|\boldsymbol{a} - \boldsymbol{x}\|_\infty$.

Lastly, suppose $a_i - (1-\gamma)\alpha \leq a_i + (1-\gamma)\alpha \leq b_i$. Then, $c_i - x_i \leq b_i - x_i \leq \|\boldsymbol{b} - \boldsymbol{x}\|_\infty \leq \gamma \|\boldsymbol{a} - \boldsymbol{x}\|_\infty$. Meanwhile, $c_i - x_i = a_i + (1-\gamma)\alpha - x_i \geq - \|\boldsymbol{a} - \boldsymbol{x}\|_\infty + (1-\gamma)\alpha$. Now, because $\|\boldsymbol{a} - \boldsymbol{x}\|_\infty \leq \alpha$, we have that $(1-\gamma)\|\boldsymbol{a} - \boldsymbol{x}\|_\infty \leq (1-\gamma)\alpha$. So, $- \|\boldsymbol{a} - \boldsymbol{x}\|_\infty + (1-\gamma)\alpha \geq \gamma \|\boldsymbol{a} - \boldsymbol{x}\|_\infty$. ∎

## B.2 Omitted proofs from Section 2

First, we prove Lemma 2.1.

**Lemma 2.1.** *Let $x = \texttt{Sample}(\boldsymbol{u}, \boldsymbol{p}, M, 0)$ for $\boldsymbol{p} \in \Delta^n$, $M \in \mathbb{Z}_{>0}$, $\varepsilon > 0$, and $\boldsymbol{u} \in \mathbb{R}^\mathcal{S}$. Then, $\mathbb{E}[x] = \boldsymbol{p}^\top \boldsymbol{u}$, $|x| \leq \|\boldsymbol{u}\|_\infty$, and $\mathrm{Var}[x] \leq 1/M \|\boldsymbol{u}\|_\infty^2$.*

*Proof.* The first statement follows from linearity of expectation and the second from definitions. The third statement follows from independence and that

$$\mathrm{Var}[v_{i_m}] = \sum_{i \in \mathcal{S}} p_i v_i^2 - (\boldsymbol{p}^\top \boldsymbol{v})^2 \leq \sum_{i \in \mathcal{S}} p_i \|\boldsymbol{v}\|_\infty^2 = \|\boldsymbol{v}\|_\infty^2 \text{ for any } m \in [M].$$

∎

Next, we state Freedman's inequality [35], which we use to prove the following Lemma 2.2.

**Theorem B.1** (Freedman's Inequality, restated from [36])**.** *Consider a real-valued martingale $\{Y_k : k = 0, 1, \ldots\}$ with difference sequence $\{X_k : k = 1, 2, \ldots\}$ given by $X_k = Y_k - Y_{k-1}$. Assume that $X_k \leq R$ almost surely for $k = 1, 2, \ldots$. Define the predictable quadratic variation process of the martingale: $W_k := \sum_{j=1}^k \mathbb{E}\left[X_j^2 | X_1, \ldots, X_{j-1}\right]$. Then, for all $t \geq 0$ and $\sigma^2 > 0$,*

$$\mathbb{P}\left\{\exists k \geq 0 : Y_k \geq t \text{ and } W_k \leq \sigma^2\right\} \leq \exp\left(-t^2/(2(\sigma^2 + Rt/3))\right)$$

**Lemma 2.2.** *Let $T \in \mathbb{Z}_{>0}$ and $\boldsymbol{w}^{(0)}, \boldsymbol{w}^{(1)}, ..., \boldsymbol{w}^{(T)} \in \mathbb{R}^{\mathcal{S}}$ such that $\left\|\boldsymbol{w}^{(i)} - \boldsymbol{w}^{(i-1)}\right\|_{\infty} \leq \tau$ for all $i \in [T]$. Then, for any $\boldsymbol{p} \in \Delta^{\mathcal{S}}$, $\delta \in (0,1)$, and $M \geq 2^8 T \log(2/\delta)$ with probability $1 - \delta$, $\left|\boldsymbol{p}^{\top}(\boldsymbol{w}^{(t)} - \boldsymbol{w}^{(0)}) - \sum_{i \in [t]} \sum_{j \in [M]} \mathtt{Sample}(\boldsymbol{w}^{(i)} - \boldsymbol{w}^{(i-1)}, \boldsymbol{p}, 1, 0) \cdot 1/M\right| \leq \tau/8$ for all $t \in [T]$.*

*Proof.* For each $i \in [T], j \in [M]$, let

$$X_{i,j} := \left(\mathtt{Sample}(\boldsymbol{w}^{(i)} - \boldsymbol{w}^{(i-1)}, \boldsymbol{p}, 1, 0) - \boldsymbol{p}^{\top}(\boldsymbol{w}^{(i)} - \boldsymbol{w}^{(i-1)})\right)/M.$$

Since $\boldsymbol{p} \in \Delta^{\mathcal{S}}$, Lemma 2.1 yields that $|X_{i,j}| \leq \frac{2\tau}{M}$. Next, define $Y_{t,k} := \sum_{i \in [t-1]} \sum_{j \in [M]} X_{i,j} + \sum_{j=1}^{k} X_{t,j}$. The predictable quadratic variation process (as defined in Theorem B.1) is given by

$$W_{t,k} = \sum_{i \in [t-1]} \sum_{j \in [M]} \mathbb{E}\left[X_{i,j}^2 | X_{1,1:M}, ..., X_{i-1,1:M}, X_{i,1:j-1}\right] + \sum_{j \in [k]} \mathbb{E}\left[X_{t,j}^2 | X_{1,1:M}, ..., X_{t-1,1:M}, X_{t,1:j-1}\right]$$

$$= \sum_{i \in [t-1]} \sum_{j \in [M]} \mathrm{Var}\left[\frac{\mathtt{Sample}(\boldsymbol{w}^{(i)} - \boldsymbol{w}^{(i-1)}, \boldsymbol{p}, 1, 0)}{M}\right] + \sum_{j \in [k]} \mathrm{Var}\left[\frac{\mathtt{Sample}(\boldsymbol{w}^{(t)} - \boldsymbol{w}^{(t-1)}, \boldsymbol{p}, 1, 0)}{M}\right]$$

$$\leq \sum_{i \in [t]} \sum_{j \in [M]} \frac{\tau^2}{M^2} = \frac{T\tau^2}{M}$$

where, in the last line we used Lemma 2.1 to bound the variance. Now, by telescoping,

$$Y_{t,M} = \left(\sum_{i \in [t]} \sum_{j \in [M]} \frac{\mathtt{Sample}(\boldsymbol{w}^{(i)} - \boldsymbol{w}^{(i-1)}, \boldsymbol{p}, 1, 0)}{M}\right) - \boldsymbol{p}^{\top}(\boldsymbol{w}^{(t)} - \boldsymbol{w}^{(0)}) \text{ for all } t \in [T]$$

Consequently, applying Theorem B.1 twice (once to $Y_{t,M}$ and once to $-Y_{t,M}$ yields

$$\mathbb{P}\left\{\exists t \in [T] : |Y_{t,M}| \geq \frac{\tau}{8}\right\} \leq 2\exp\left(-\frac{(\tau/8)^2}{2(\frac{T\tau^2}{M} + \frac{2\tau}{M} \cdot \frac{\tau}{8} \cdot \frac{1}{3})}\right) = 2\exp\left(\frac{-M}{2^7(T + \frac{1}{12})}\right) \leq \delta.$$

∎

As an immediate corollary of Lemma 2.2, we obtain Corollary 2.3.

**Corollary 2.3.** *In* `TVRVI` *(Algorithm 3), with probability $1 - \delta$, in Lines 9, 10 and 2, for all $s \in \mathcal{S}, a \in \mathcal{A}_s$, and $\ell \in [L]$, we have $\left|\boldsymbol{g}_a^{(\ell)}(s) - \boldsymbol{p}_a(s)^{\top}(\boldsymbol{v}^{(\ell-1)} - \boldsymbol{v}^{(0)})\right| \leq (1-\gamma)\alpha/8$ and therefore $\hat{\boldsymbol{g}}_a^{(\ell)}$ is a $(1-\gamma)\alpha/4$-underestimate of $\boldsymbol{p}_a(s)^{\top}(\boldsymbol{v}^{(\ell-1)} - \boldsymbol{v}^{(0)})$.*

*Proof.* Consider some $s \in \mathcal{S}$ and $a \in \mathcal{A}_s$. Note that $\boldsymbol{g}_a^{(\ell)}(s)$ is equal in distribution to

$$\left(\sum_{i \in [\ell-1]} \sum_{j \in [M]} \frac{\mathtt{Sample}(\boldsymbol{v}^{(i)} - \boldsymbol{v}^{(i-1)}, \boldsymbol{p}_a(s), 1, 0)}{M}\right) - \boldsymbol{p}_a(s)^{\top}(\boldsymbol{v}^{(\ell-1)} - \boldsymbol{v}^{(0)}).$$

Then, by Lemma 2.2 and union bound, whenever $M \geq L \cdot 2^8 \log(2\mathcal{A}_{\mathrm{tot}}/\delta)$ we have that with probability $1 - \delta$, for all $(s, a) \in \mathcal{A}$, $\left|\boldsymbol{g}_a^{(\ell)}(s) - \boldsymbol{p}_a(s)^{\top}(\boldsymbol{v}^{(\ell-1)} - \boldsymbol{v}^{(0)})\right| \leq \frac{1-\gamma}{8}\alpha$ and conditioning on this event, we have $\boldsymbol{p}_a(s)^{\top}(\boldsymbol{v}^{(\ell-1)} - \boldsymbol{v}^{(0)}) - \frac{1-\gamma}{4}\alpha \leq \hat{\boldsymbol{g}}_a^{(\ell)}(s) \leq \boldsymbol{p}_a(s)^{\top}(\boldsymbol{v}^{(\ell-1)} - \boldsymbol{v}^{(0)})$ due to the shift in Line 10. ∎

Conditioning on the event that the implication of Corollary 2.3 holds, we can prove the following Lemma 2.4

**Lemma 2.4.** *Suppose that for some $\boldsymbol{\beta} \in \mathbb{R}_{\geq 0}^{\mathcal{A}}$, $\boldsymbol{P}\boldsymbol{v}^{(0)} - \boldsymbol{\beta} \leq \boldsymbol{x} \leq \boldsymbol{P}\boldsymbol{v}^{(0)}$ and let $\boldsymbol{\beta}_{\pi^{\star}} \in \mathbb{R}^{\mathcal{S}}$ be defined as $\boldsymbol{\beta}_{\pi^{\star}}(s) := \boldsymbol{\beta}_{\pi^{\star}(s)}(s)$ for each $s \in \mathcal{S}$. Then, with probability $1 - \delta$, at the end of every iteration $\ell \in [L]$ (Line 3) in* `TVRVI`$(\boldsymbol{v}^{(0)}, \pi^{(0)}, \boldsymbol{x}, \alpha, \delta)$, *the following hold for $\boldsymbol{\xi} := \gamma((1-\gamma)\alpha/4\mathbf{1} + \boldsymbol{\beta}_{\pi^{\star}})$:*

$$\boldsymbol{v}^{(\ell-1)} \leq \boldsymbol{v}^{(\ell)} \leq \mathcal{T}_{\pi^{(\ell)}}(\boldsymbol{v}^{(\ell)}), \tag{6}$$

$$0 \leq \boldsymbol{v}^{\star} - \boldsymbol{v}^{(\ell)} \leq \max\left(\gamma\boldsymbol{P}^{\star}(\boldsymbol{v}^{\star} - \boldsymbol{v}^{(\ell-1)}) + \boldsymbol{\xi}, \gamma(\boldsymbol{v}^{\star} - \boldsymbol{v}^{(\ell-1)})\right). \tag{7}$$

*Proof.* In the remainder of this proof, condition on the event that the implications of Corollary 2.3 hold (as they occur with probability $1 - \delta$). By Line 7 and 8 of Algorithm 3, for all $\ell \in [L]$,

$$\boldsymbol{v}^{(\ell-1)} \leq \boldsymbol{v}^{(\ell)} \leq \boldsymbol{v}^{(\ell-1)} + (1-\gamma)\alpha\mathbf{1}.$$

This immediately implies the lower bound in (6).

We prove the upper bound in (6) by induction. In the base case when $\ell = 0$, $\boldsymbol{v}^{(0)} \leq \mathcal{T}_{\pi^{(0)}}(\boldsymbol{v}^{(0)})$ holds by assumption. For the $\ell$-th iteration, there are two cases. If $\boldsymbol{v}^{(\ell)}(s) > \boldsymbol{v}^{(\ell-1)}(s)$ for $s \in S$ then

$$\boldsymbol{v}^{(\ell)}(s) = \boldsymbol{r}_{\pi^{(\ell)}}(s) + \gamma\left(\boldsymbol{x}(s) + \hat{\boldsymbol{g}}^{(\ell)}_{\pi^{(\ell)}}(s)\right) \leq \boldsymbol{r}_{\pi^{(\ell)}}(s) + \gamma\boldsymbol{p}_{\pi^{(\ell)}}(s)^\top\boldsymbol{v}^{(\ell-1)}(s) \qquad (10)$$
$$\leq \mathcal{T}_{\pi^{(\ell)}}(\boldsymbol{v}^{(\ell-1)}) \leq \mathcal{T}_{\pi^{(\ell)}}(\boldsymbol{v}^{(\ell)}).$$

Otherwise, if $\boldsymbol{v}^{(\ell)}(s) = \boldsymbol{v}^{(\ell-1)}(s)$, then by the inductive hypothesis,

$$\boldsymbol{v}^{(\ell)}(s) = \boldsymbol{v}^{(\ell-1)}(s) \leq \mathcal{T}_{\pi^{(\ell-1)}}(\boldsymbol{v}^{(\ell-1)})(s) = \mathcal{T}_{\pi^{(\ell)}}(\boldsymbol{v}^{(\ell)})(s)\,.$$

This completes the proof of (6).

Next, we prove (7). For the lower bound, by induction and (10), we have that for each $s \in \mathcal{S}$

$$\tilde{\boldsymbol{v}}^{(\ell)}(s) \leq \max_{a \in \mathcal{A}_s}\{\boldsymbol{r}_a(s) + \gamma\boldsymbol{p}_a(s)^\top\boldsymbol{v}^{(\ell-1)}(s)\} \leq \max_{a \in \mathcal{A}_s}\{\boldsymbol{r}_a(s) + \gamma\boldsymbol{p}_a(s)^\top\boldsymbol{v}^\star(s)\} = \boldsymbol{v}^\star,$$

so $\min(\tilde{\boldsymbol{v}}^{(\ell)}, \boldsymbol{v}^{(\ell-1)} + (1-\gamma)\alpha) \leq \boldsymbol{v}^\star$.

Next, we prove the upper bound of (7). For each $(s, a) \in \mathcal{A}$ and $\ell \in [L]$, let

$$\boldsymbol{\xi}^{(\ell)}_a(s) := \boldsymbol{p}_a(s)^\top\boldsymbol{v}^{(\ell-1)} - (\boldsymbol{x}_a(s) + \hat{\boldsymbol{g}}^{(\ell)}_a)(s)),$$

and observe that

$$\boldsymbol{\xi}^{(\ell)}_a(s) = [\boldsymbol{p}_a(s)^\top\boldsymbol{v}^{(0)} - \boldsymbol{x}_a(s)] + [\boldsymbol{p}_a(s)^\top(\boldsymbol{v}^{(\ell-1)} - \boldsymbol{v}^{(0)}) - \hat{\boldsymbol{g}}^{(\ell)}_a)(s))] \leq \boldsymbol{\beta}_a(s) + \frac{(1-\gamma)\alpha}{4}.$$

Note that for any $s \in \mathcal{S}$,

$$(\boldsymbol{v}^\star - \tilde{\boldsymbol{v}}^{(\ell)})(s) = \max_{a \in \mathcal{A}_i}[\boldsymbol{r}_a(s) + \gamma\boldsymbol{p}_a(s)^\top\boldsymbol{v}^\star(s)] - \max_{a \in \mathcal{A}_s}[\boldsymbol{r}_a(s) + \gamma(\boldsymbol{x}_a(s) + \hat{\boldsymbol{g}}^{(\ell)}_a)(s))]$$
$$\leq [\boldsymbol{r}_{\pi^\star(s)}(s) + \gamma\left(\boldsymbol{P}^\star\boldsymbol{v}^\star\right)(s)] - \max_{a \in \mathcal{A}_s}[\boldsymbol{r}_a(s) + \gamma\boldsymbol{p}_a(s)^\top\boldsymbol{v}^{(\ell-1)} - \gamma\boldsymbol{\xi}^{(\ell)}_a(s)]$$
$$\leq [\boldsymbol{r}_{\pi^\star(s)}(s) + \gamma\left(\boldsymbol{P}^\star\boldsymbol{v}^\star\right)(s)] - [\boldsymbol{r}_{\pi^\star(s)}(s) + \gamma(\boldsymbol{P}^\star\boldsymbol{v}^{(\ell-1)})(s) - \gamma\boldsymbol{\xi}^{(\ell)}_{\pi^\star(s)}(s)]$$
$$\leq \gamma\left(\boldsymbol{P}^\star(\boldsymbol{v}^\star - \boldsymbol{v}^{(\ell-1)})\right)(s) + \boldsymbol{\xi}(s),$$

Consequently, for all $s \in S$,

$$(\boldsymbol{v}^\star - \tilde{\boldsymbol{v}}^{(\ell)})(s) \leq \gamma\boldsymbol{P}^\star(\boldsymbol{v}^\star - \boldsymbol{v}^{(\ell-1)})(s) + \boldsymbol{\xi}(s).$$

Consider two cases for $\boldsymbol{v}^{(\ell)}(s)$. First, if $\boldsymbol{v}^{(\ell)}(s) = \tilde{\boldsymbol{v}}^{(\ell)}(s)$ for some $s \in \mathcal{S}$ then

$$\left(\boldsymbol{v}^\star - \boldsymbol{v}^{(\ell)}\right)(s) \leq \gamma\left(\boldsymbol{P}^\star\left(\boldsymbol{v}^\star - \boldsymbol{v}^{(\ell-1)}\right)\right)(s) + \boldsymbol{\xi}(s)$$

holds immediately. If not, $\boldsymbol{v}^{(\ell)}(s) = \boldsymbol{v}^{(\ell-1)}(s) + (1-\gamma)\alpha \leq \tilde{\boldsymbol{v}}^{(\ell)}(s)$ and (6) guarantees that

$$\left\|\boldsymbol{v}^\star - \boldsymbol{v}^{(\ell-1)}\right\|_\infty \leq \left\|\boldsymbol{v}^\star - \boldsymbol{v}^{(0)}\right\|_\infty \leq \alpha,$$

which ensures that $(1-\gamma)(\boldsymbol{v}^\star - \boldsymbol{v}^{(\ell-1)})(s) \leq (1-\gamma)\alpha$ and yields the results as,

$$\left(\boldsymbol{v}^\star - \boldsymbol{v}^{(\ell)}\right)(s) = \left(\boldsymbol{v}^\star - \boldsymbol{v}^{(\ell-1)}\right)(s) - (1-\gamma)\alpha \leq \gamma\left(\boldsymbol{v}^\star - \boldsymbol{v}^{(\ell-1)}\right)(s)\,.$$

■

We now inductively apply Lemma 2.4 to obtain Corollary 2.5, which allows us to bound the number of iterates required to halve the initial error in TVRVI.

**Corollary 2.5.** *Suppose that for some $\alpha \geq 0$ and $\boldsymbol{\beta} \in \mathbb{R}_{\geq 0}^{\mathcal{A}}$, $\boldsymbol{P}\boldsymbol{v}^{(0)} - \boldsymbol{\beta} \leq \boldsymbol{x} \leq \boldsymbol{P}\boldsymbol{v}^{(0)}$; $\boldsymbol{v}^{(0)}$ is an $\alpha$-underestimate of $\boldsymbol{v}^\star$; and $\boldsymbol{v}^{(0)} \leq \mathcal{T}_{\pi^{(0)}}(\boldsymbol{v}^{(0)})$. Let $\boldsymbol{\beta}_{\pi^\star} \in \mathbb{R}^{\mathcal{S}}$ be defined as $\boldsymbol{\beta}_{\pi^\star}(s) := \boldsymbol{\beta}_{\pi^\star(s)}(s)$ for each $s \in \mathcal{S}$. Let $(\boldsymbol{v}^{(L)}, \pi^{(L)}) = \mathtt{TVRVI}(\boldsymbol{v}^{(0)}, \pi^{(0)}, \alpha, \delta)$, and $L, M$ be as in Line 2. Define $\boldsymbol{\xi} := \gamma\left((1-\gamma)\alpha/4 \cdot \boldsymbol{1} + \boldsymbol{\beta}_{\pi^\star}\right)$. Then, with probability $1 - \delta$, $\boldsymbol{0} \leq \boldsymbol{v}^\star - \boldsymbol{v}^{(L)} \leq \gamma^L\alpha \cdot \boldsymbol{1} + (\boldsymbol{I} - \gamma\boldsymbol{P}^\star)^{-1}\boldsymbol{\xi}$, and $\boldsymbol{v}^{(L)} \leq \mathcal{T}_{\pi^{(L)}}(\boldsymbol{v}^{(L)})$. In particular, if $\boldsymbol{\beta} = \boldsymbol{0}$, then for $L > \log(8)(1-\gamma)^{-1}$ we can reduce the error in $\boldsymbol{v}^{(0)}$ by half: $\boldsymbol{0} \leq \boldsymbol{v}^\star - \boldsymbol{v}^{(L)} \leq (\boldsymbol{v}^\star - \boldsymbol{v}^{(0)})/2$. Additionally, $\mathtt{TVRVI}$ is implementable with $\tilde{O}(\mathcal{A}_{\mathrm{tot}}ML)$ sample queries to the generative model and time and $O(\mathcal{A}_{\mathrm{tot}})$ space.*

*Proof.* Condition on the event that the implication of Lemma 2.4 holds. First, we observe that $\boldsymbol{0} \leq \boldsymbol{v}^\star - \boldsymbol{v}_{\pi^{(L)}} \leq \boldsymbol{v}^\star - \boldsymbol{v}^{(L)}$ follows by monotonicity (Equation (6) of Lemma 2.4). Next, we show that

$$\boldsymbol{v}^\star - \boldsymbol{v}^{(L)} \leq \gamma^L\alpha \cdot \boldsymbol{1} + (\boldsymbol{I} - \gamma\boldsymbol{P}^\star)^{-1}\boldsymbol{\xi},$$

by induction. We will show that for all $i \in \mathcal{S}$,

$$\boldsymbol{v}^\star - \boldsymbol{v}^{(\ell)} \leq \left[\gamma^\ell\alpha\boldsymbol{1} + \sum_{k=0}^{\ell}\gamma^k\boldsymbol{P}^{\star k}\boldsymbol{\xi}\right].$$

In the base case when $\ell = 0$, this is trivially true, as $\boldsymbol{v}^\star - \boldsymbol{v}^{(\ell)} \leq \alpha\boldsymbol{1}$ by assumption. Assume that the statement is true up to $\boldsymbol{v}^{(\ell-1)}$. Now, by Lemma 2.4, we have two cases for $[\boldsymbol{v}^\star - \boldsymbol{v}^{(\ell)}](i)$.

First, suppose that $[\boldsymbol{v}^\star - \boldsymbol{v}^{(\ell)}](i) \leq \gamma[\boldsymbol{v}^\star - \boldsymbol{v}^{(\ell-1)}](i)$. Then, note that $\boldsymbol{P}^\star$ and $\boldsymbol{\xi}$ are entrywise non-negative, so $[\gamma^\ell\boldsymbol{P}^{\star\ell}\boldsymbol{\xi}](i) \geq 0$. By inductive hypothesis, and the fact that $\gamma \in (0,1)$ we have

$$[\boldsymbol{v}^\star - \boldsymbol{v}^{(\ell)}](i) \leq \gamma\left(\gamma^{(\ell-1)}\alpha + \left[\sum_{k=0}^{\ell-1}\gamma^k\boldsymbol{P}^{\star k}\boldsymbol{\xi}\right](i)\right)$$

$$= \gamma^\ell\alpha + \gamma\left[\sum_{k=0}^{\ell-1}\gamma^k\boldsymbol{P}^{\star k}\boldsymbol{\xi}\right](i) \leq \gamma^\ell\alpha + \left[\sum_{k=0}^{\ell-1}\gamma^k\boldsymbol{P}^{\star k}\boldsymbol{\xi}\right](i) \leq \gamma^\ell\alpha + \left[\sum_{k=0}^{\ell}\gamma^k\boldsymbol{P}^{\star k}\boldsymbol{\xi}\right](i)$$

$$= \left[\gamma^\ell\alpha\boldsymbol{1} + \sum_{k=0}^{\ell}\gamma^k\boldsymbol{P}^{\star k}\boldsymbol{\xi}\right](i).$$

Second, suppose that instead, $[\boldsymbol{v}^\star - \boldsymbol{v}^{(\ell)}](i) \leq \left[\gamma\boldsymbol{P}^\star\left(\boldsymbol{v}^\star - \boldsymbol{v}^{(\ell-1)}\right)\right](i) + \boldsymbol{\xi}(i)$. By monotonicity (equation (6) of Lemma 2.4) we know that $\boldsymbol{v}^\star - \boldsymbol{v}^{(\ell-1)} \geq 0$. Moreover, $\boldsymbol{P}^\star$ is non-negative, and consequently, we can use the inductive hypothesis as follows:

$$\left(\boldsymbol{v}^\star - \boldsymbol{v}^{(\ell-1)}\right) \leq \left[\gamma^{\ell-1}\alpha\boldsymbol{1} + \sum_{k=0}^{\ell-1}\gamma^k\boldsymbol{P}^{\star k}\boldsymbol{\xi}\right], \text{ hence } \boldsymbol{P}^\star\left(\boldsymbol{v}^\star - \boldsymbol{v}^{(\ell-1)}\right) \leq \boldsymbol{P}^\star\left[\gamma^{\ell-1}\alpha\boldsymbol{1} + \sum_{k=0}^{\ell-1}\gamma^k\boldsymbol{P}^{\star k}\boldsymbol{\xi}\right].$$

We can rearrange terms to obtain the following bound:

$$[\boldsymbol{v}^\star - \boldsymbol{v}^{(\ell)}](i) \leq \left[\gamma\boldsymbol{P}^\star\left(\gamma^{(\ell-1)}\alpha\boldsymbol{1} + \sum_{k=0}^{\ell-1}\gamma^k\boldsymbol{P}^{\star k}\boldsymbol{\xi}\right)\right](i) + \boldsymbol{\xi}(i)$$

$$= \gamma^\ell\alpha[\boldsymbol{P}^\star\boldsymbol{1}](i) + \left[\sum_{k=0}^{\ell-1}\gamma^{k+1}\boldsymbol{P}^{\star k+1}\boldsymbol{\xi}\right](i) + \boldsymbol{\xi}(i) \leq \left[\gamma^\ell\alpha\boldsymbol{1} + \sum_{k=0}^{\ell}\gamma^k\boldsymbol{P}^{\star k}\boldsymbol{\xi}\right](i).$$

Consequently, by induction, the bound holds. When $L > \log(8)(1-\gamma)^{-1}$, $\gamma^L \leq 1/8$ and we have

$$\boldsymbol{v}^\star - \boldsymbol{v}_k \leq \gamma^L\alpha \cdot \boldsymbol{1} + (\boldsymbol{I} - \gamma\boldsymbol{P}^\star)^{-1}\frac{\gamma(1-\gamma)}{4}\alpha\boldsymbol{1} \leq \gamma^L\alpha + \gamma\frac{\alpha}{4} \leq \frac{\alpha}{2}.$$

Finally, the sample complexity and runtime follow from the algorithm pseudocode. For the space complexity, at each iteration $\ell$ of the outer for loop in $\mathtt{TVRVI}$, the algorithm needs only to maintain $\hat{\boldsymbol{g}}^{(\ell)}, \boldsymbol{g}^{(\ell)} \in \mathbb{R}^{\mathcal{A}_{\mathrm{tot}}}$, $\boldsymbol{v}^{(\ell)} \in \mathbb{R}^S$, $\pi^{(L)}$, and at most $M\mathcal{A}_{\mathrm{tot}}$ samples in invoking $\mathtt{Sample}$.

∎

Finally, we are ready to prove Theorem 1.2.

**Theorem 1.2.** *In the offline setting, there is an algorithm that uses $\tilde{O}(\mathrm{nnz}(\boldsymbol{P}) + \mathcal{A}_{\mathrm{tot}}(1-\gamma)^{-2})$ time, and computes an $\varepsilon$-optimal policy and $\varepsilon$-optimal values with probability $1 - \delta$.*

*Proof.* To run `OfflineTVRVI`, we can implement a generative model from which we can draw samples in $O(\mathrm{nnz}(\boldsymbol{P}))$ pre-processing time, so that each query to the generative model requires $\tilde{O}(1)$ time. For the correctness, we induct on $k$ to show that after each iteration $k$, $0 \leq \boldsymbol{v}^\star - \boldsymbol{v}_{\pi_K} \leq \boldsymbol{v}^\star - \boldsymbol{v}_K \leq \alpha_k$ with probability $1 - k\delta/K$. In the base case when $k = 0$, the bound is trivially true as $\|\boldsymbol{v}^\star\|_\infty \leq (1-\gamma)^{-1}$. Now, by Applying Corollary 2.5 and a union bound, we see that with probability $1 - k\delta/K$, $\boldsymbol{v}^\star - \boldsymbol{v}_k \leq \frac{\alpha_{k-1}}{2} = \alpha_k$, whenever $L > \log(8)(1-\gamma)^{-1}$. Thus, $\boldsymbol{v}_K$ satisfies the required guarantee whenever $\alpha_K \leq \varepsilon$, which is guaranteed by our choice of $K$. To see that $\pi_k$ is an $\varepsilon$-optimal policy, we observe that Corollary 2.5 ensures

$$\boldsymbol{v}_k \leq \mathcal{T}_{\pi_k}(\boldsymbol{v}_k) \leq \mathcal{T}_{\pi_k}^2(\boldsymbol{v}_k) \leq \cdots \leq \mathcal{T}_{\pi_k}^\infty(\boldsymbol{v}_k) = \boldsymbol{v}^{\pi_k} \leq \boldsymbol{v}^\star.$$

For the runtime, the algorithm completes only $K = \tilde{O}(1)$ iterations, and can be implemented with $\tilde{O}(1)$ calls to the offset oracle. Each inner loop iteration can be implemented with $\tilde{O}(\mathcal{A}_{\mathrm{tot}}L^2) = \tilde{O}\left(\mathcal{A}_{\mathrm{tot}}(1-\gamma)^{-2}\right)$ additional time and queries to the generative model. The algorithm only requires $O(\mathcal{A}_{\mathrm{tot}})$ space in order to store offsets, values, and approximate utilities. ∎

## B.3 Omitted proofs from Section 3

**Theorem B.2** (Hoeffding's Inequality and Bernstein's Inequality, restated from Lemma E.1 and E.2 of [3]). *Let $\boldsymbol{p} \in \Delta^{\mathcal{S}}$ be a probability vector, $\boldsymbol{v} \in \mathbb{R}^n$, and let $\boldsymbol{y} := \frac{1}{m}\sum_{j=1}^m \boldsymbol{v}(i_j)$ where $i_j$ are random indices drawn such that $i_j = k$ with probability $\boldsymbol{p}(k)$. Define $\sigma := (\boldsymbol{p}^\top \boldsymbol{v}^2 - (\boldsymbol{p}^\top \boldsymbol{v})^2)$. For any $\delta \in (0,1)$, the following hold, each with probability $1 - \delta$:*

(Hoeffding's Inequality) $\left|\boldsymbol{p}^\top \boldsymbol{v} - \boldsymbol{y}\right| \leq \|\boldsymbol{v}\|_\infty \cdot \sqrt{2m^{-1}\log(2\delta^{-1})}$,

(Bernstein's Inequality) $\left|\boldsymbol{p}^\top \boldsymbol{v} - \boldsymbol{y}\right| \leq \sqrt{2m^{-1}\sigma \cdot \log(2\delta^{-1})} + (2/3)m^{-1}\|\boldsymbol{v}\|_\infty \cdot \log(2\delta^{-1})$.

Theorem B.2 illustrates that the error in estimating $\boldsymbol{P}\boldsymbol{u}$ for some value vector $\boldsymbol{u}$ depends on the variance $\boldsymbol{\sigma}_{\boldsymbol{u}} := \boldsymbol{P}\boldsymbol{u}^2 - (\boldsymbol{P}\boldsymbol{u})^2 \in \mathbb{R}^{\mathcal{A}}$. To bound this variance term, we appeal to the following two lemmas from [3].

**Lemma B.3** (Lemma 5.2 of ([3]), restated). $\sqrt{\boldsymbol{\sigma}_{\boldsymbol{v}}} \leq \sqrt{\boldsymbol{\sigma}_{\boldsymbol{v}^\star}} + \|\boldsymbol{v}^\star - \boldsymbol{v}\|_\infty \mathbf{1}$.

**Lemma B.4** (Lemma C.1 of ([3]), restated). *For any $\pi$, we have*

$$\left\|(\boldsymbol{I} - \gamma\boldsymbol{P}^\pi)^{-1}\sqrt{\boldsymbol{\sigma}_{\boldsymbol{v}^\pi}}\right\|_\infty^2 \leq \frac{1+\gamma}{\gamma^2(1-\gamma)^3}.$$

We can now bound the error in estimating $\boldsymbol{P}\boldsymbol{u}$ using `ApxUtility`$(\boldsymbol{u}, N, \eta)$. The following Lemma 3.1 obtains such a bound by following a similar argument to that of Lemma 5.1 of [3].

**Lemma 3.1.** *Consider $\boldsymbol{u} \in \mathbb{R}^{\mathcal{S}}$. Let $\boldsymbol{x} = $ `ApxUtility`$(\boldsymbol{u}, m \cdot \mathcal{A}_{\mathrm{tot}}, \eta)$, $m \geq \log(1/2\delta^{-1})$, and $\eta = (m\mathcal{A}_{\mathrm{tot}})^{-1}\log(1/2\delta^{-1})$. Then, with probability $1 - \delta$,*

$$\boldsymbol{P}\boldsymbol{u} - 2\sqrt{2\eta\boldsymbol{\sigma}_{\boldsymbol{v}^\star}} + \left(2\sqrt{2\eta}\|\boldsymbol{u} - \boldsymbol{v}^\star\|_\infty + 18\eta^{3/4}\|\boldsymbol{u}\|_\infty\right) \leq \boldsymbol{x} \leq \boldsymbol{P}\boldsymbol{u}.$$

*Proof.* For $s \in \mathcal{S}$ and $a \in \mathcal{A}_s$. Let $i_1, ..., i_N \in \mathcal{S}$ be random indices such that $\mathbb{P}\{i_j = t\} = (\boldsymbol{p}_a(s))(t)$ for each $j \in [N]$. Define the vectors $\tilde{\boldsymbol{x}}$ and $\hat{\boldsymbol{\sigma}}$ as follows.

$$\tilde{\boldsymbol{x}}_a(s) := \frac{1}{N}\sum_{j=1}^N \boldsymbol{u}(i_j) \text{ and } \hat{\boldsymbol{\sigma}}_a(s) := \frac{1}{N}\sum_{j=1}^N (\boldsymbol{u}(i_j))^2 - (\tilde{\boldsymbol{x}}_a(s))^2.$$

From the pseudocode of `ApxUtility` (Algorithm 1), we see that that $\boldsymbol{x} = \tilde{\boldsymbol{x}} - \sqrt{2\eta\hat{\boldsymbol{\sigma}}} - 4\eta^{3/4}\|\boldsymbol{u}\|_\infty - (2/3)\eta\|\boldsymbol{u}\|_\infty$. Now, by union bound over all state-action pairs $(s, a)$ and Theorem B.2, we have that with probability $1 - \delta/2$ for each sate-action pair $(s, a)$,

$$\left\| \boldsymbol{x} - \boldsymbol{Pu}_\infty \le \sqrt{2\eta\boldsymbol{\sigma_u}} \right\| + \frac{2}{3}\eta\|\boldsymbol{u}\|_\infty \mathbf{1}. \tag{11}$$

and with probability $1 - \delta/2$ for each sate-action pair $(s, a)$,

$$\left\| \frac{1}{N} \sum_{j \in [N]} (\hat{\boldsymbol{\sigma}}_a(s))^2 - \boldsymbol{p}_a(s)^\top \boldsymbol{u}^2 \right\| \le \|\boldsymbol{u}\|_\infty^2 \sqrt{2\eta}_\infty.$$

Consequently, by union bound and triangle inequality and (11), we have that with probability $1 - \delta$ both of the following hold.

$$\|\tilde{\boldsymbol{x}} - \boldsymbol{Pu}\|_\infty \le \sqrt{2\eta\boldsymbol{\sigma_u}} + \frac{2}{3}\eta\|\boldsymbol{u}\|_\infty \mathbf{1}, \text{ and } \|\hat{\boldsymbol{\sigma}} - \boldsymbol{\sigma_u}\|_\infty \le 4\|\boldsymbol{u}\|_\infty^2 \cdot \sqrt{2\eta}\mathbf{1}. \tag{12}$$

We condition on (12) in the remainder of the proof. Now,

$$|\tilde{\boldsymbol{x}} - \boldsymbol{Pu}| \le \sqrt{2\eta\hat{\boldsymbol{\sigma}}} + \left( 4\eta^{3/4}\|\boldsymbol{u}\|_\infty + \frac{2}{3}\eta\|\boldsymbol{u}\|_\infty \right) \mathbf{1},$$

and we have that

$$\boldsymbol{Pu} - 2\sqrt{2\eta\hat{\boldsymbol{\sigma}}} - \left( 8\eta^{3/4}\|\boldsymbol{u}\|_\infty + \frac{4}{3}\eta\|\boldsymbol{u}\|_\infty \right) \mathbf{1} \le \boldsymbol{x} \le \boldsymbol{Pu}.$$

By (12) and Lemma B.3, we have that for $\alpha := \|\boldsymbol{u} - \boldsymbol{v}^\star\|_\infty$,

$$\sqrt{\hat{\boldsymbol{\sigma}}} \le \sqrt{\boldsymbol{\sigma_u}} + 2\|\boldsymbol{u}\|_\infty (2\eta)^{1/4}\mathbf{1} \le \sqrt{\boldsymbol{\sigma_{v^\star}}} + \alpha\mathbf{1} + 2\|\boldsymbol{u}\|_\infty (2\eta)^{1/4}\mathbf{1},$$

which implies that

$$\boldsymbol{x} \ge \boldsymbol{Pu} - 2\sqrt{2\eta\boldsymbol{\sigma_{v^\star}}} - 2\sqrt{2\eta}\alpha\mathbf{1} - 16\eta^{3/4}\|\boldsymbol{u}\|_\infty \mathbf{1} - \frac{4}{3}\eta\|\boldsymbol{u}\|_\infty \mathbf{1}.$$

Since $\eta \le 1$,

$$2\sqrt{2\eta\boldsymbol{\sigma_{v^\star}}} + \left( 2\sqrt{2\eta}\alpha + 16\eta^{3/4}\|\boldsymbol{u}\|_\infty + \frac{4}{3}\eta\|\boldsymbol{u}\|_\infty \right) \mathbf{1} \le 2\sqrt{2\eta\boldsymbol{\sigma_{v^\star}}} + \left( 2\sqrt{2\eta}\alpha + 18\eta^{3/4}\|\boldsymbol{u}\|_\infty \right) \mathbf{1}.$$

∎

**Theorem 1.1.** *In the sample setting, there is an algorithm that uses $\tilde{O}(\mathcal{A}_{\text{tot}}[(1-\gamma)^{-3}\varepsilon^{-2} + (1-\gamma)^{-2}])$ samples and time and $O(\mathcal{A}_{\text{tot}})$ space, and computes an $\varepsilon$-optimal policy and $\varepsilon$-optimal values with probability $1 - \delta$.*

*Proof.* Let $K$, $\alpha_k$, $(\boldsymbol{v}_k, \pi_k)$, and $N_k$ be as defined in Lines 1, 4, 9, and 6 of `SampleTVRVI`$(\varepsilon, \delta)$. First, we show, by induction that for each $k \in [K]$, with probability $1 - k\delta/K$,

$$\boldsymbol{0} \le \boldsymbol{v}^\star - \boldsymbol{v}^{\pi_k} \le \boldsymbol{v}^\star - \boldsymbol{v_k} \le \alpha_k\mathbf{1} \text{ and } \boldsymbol{v_k} \le \mathcal{T}_{\pi_k}(\boldsymbol{v_k}).$$

In the base case when $k = 0$, the bound is trivially true because $\boldsymbol{0} \le \boldsymbol{v}^\star - \boldsymbol{v}_{\pi_0} \le \boldsymbol{v}^\star - \boldsymbol{v_0} \le (1-\gamma)^{-1}$.

Now, for the inductive step, by Lemma 3.1 we see that with probability $1 - \delta/K$,

$$\boldsymbol{Pv}_{k-1} - \left[ 2\sqrt{2\eta_{k-1}\boldsymbol{\sigma_{v^\star}}} + \left( 2\sqrt{2\eta_{k-1}}\alpha_{k-1} + 18\eta_{k-1}^{3/4}\|\boldsymbol{v}_{k-1}\|_\infty \right) \mathbf{1} \right] \le \boldsymbol{x_k} \le \boldsymbol{Pv}_{k-1} \tag{13}$$

and, by inductive hypothesis, with probability $1 - (k-1)\delta/K$,

$$\boldsymbol{0} \le \boldsymbol{v}^\star - \boldsymbol{v}^{\pi_{k-1}} \le \boldsymbol{v}^\star - \boldsymbol{v}_{k-1} \le \alpha_{k-1}\mathbf{1}, \text{ and } \boldsymbol{v_k} \le \mathcal{T}_{\pi_{k-1}}(\boldsymbol{v}_{k-1}). \tag{14}$$

Consequently, by a union bound, with probability $1 - k\delta/K$, both (13) and (14) hold. Condition on this event for the remainder of the inductive step.

Next, we can apply Corollary 2.5 with

$$\boldsymbol{\beta} = 2\sqrt{2\eta_{k-1}}\boldsymbol{\sigma}_{\boldsymbol{v}^\star} + \left(2\sqrt{2\eta_{k-1}}\alpha_{k-1} + 18\eta_{k-1}^{3/4}\|\boldsymbol{v}_{k-1}\|_\infty\right)\mathbf{1}.$$

Therefore,

$$0 \le \boldsymbol{v}^\star - \boldsymbol{v}_k \le \gamma^L\alpha_{k-1}\cdot\mathbf{1} + (\boldsymbol{I} - \gamma\boldsymbol{P}^\star)^{-1}\boldsymbol{\xi}_{k-1} \le \frac{\alpha_{k-1}}{8}\mathbf{1} + (\boldsymbol{I} - \gamma\boldsymbol{P}^\star)^{-1}\boldsymbol{\xi}_{k-1}$$

for $\boldsymbol{\xi}_{k-1} \le \frac{(1-\gamma)\alpha_{k-1}}{4}\mathbf{1} + 2\sqrt{2\eta_{k-1}}\boldsymbol{\sigma}_{\boldsymbol{v}^\star} + \left(2\sqrt{2\eta_{k-1}}\alpha_{k-1} + 18\eta_{k-1}^{3/4}\|\boldsymbol{v}_{k-1}\|_\infty\right)\mathbf{1}$. By Lemma B.4 and the facts that $\eta_{k-1} \le (6500\cdot(1-\gamma)^{-3}\max((1-\gamma),\alpha_{k-1}^{-2}))^{-1}$ and $(\boldsymbol{I}-\gamma\boldsymbol{P}^\star)^{-1}\mathbf{1} = 1/(1-\gamma)\mathbf{1}$, we obtain

$$(\boldsymbol{I} - \gamma\boldsymbol{P}^\star)^{-1}\boldsymbol{\xi}_{k-1} \le \left[\frac{\alpha_{k-1}}{4} + 2\sqrt{\frac{6\eta_{k-1}}{(1-\gamma)^3}} + 2\sqrt{\frac{2(1-\gamma)^3\min((1-\gamma)^{-1},\alpha_{k-1}^2)}{6500(1-\gamma)^2}}\alpha_{k-1}\right]\mathbf{1}$$

$$+ \left[18\left(\frac{((1-\gamma)^3\min((1-\gamma)^{-1},\alpha_{k-1}^2))}{6500(1-\gamma)^{8/3}}\right)^{3/4}\right]\mathbf{1}$$

$$\le [\alpha_{k-1}/4 + 2\sqrt{6/6500}\cdot\alpha_{k-1} + 2\sqrt{2/6500}(1-\gamma)^{1/2}\min((1-\gamma)^{-1/2},\alpha_{k-1})\alpha_{k-1}$$

$$+ 18\cdot(10^{-3})(1-\gamma)^{1/4}\min((1-\gamma)^{-3/4},\alpha_{k-1}^{3/2})]\mathbf{1}$$

$$\le [\alpha_{k-1}/4 + 4\sqrt{6/6500}\cdot\alpha_{k-1} + 18\cdot(10^{-3})\alpha_{k-1}]\mathbf{1} \le \frac{3}{8}\alpha_{k-1}\mathbf{1}.$$

Consequently, $\boldsymbol{v}^\star - \boldsymbol{v}_k \le \alpha/2\mathbf{1}$. To see that $\pi_k$ is also an $\alpha_k$-optimal policy, we observe that Corollary 2.5 also ensures that

$$\boldsymbol{v}_k \le \mathcal{T}_{\pi_k}(\boldsymbol{v}_k) \le \mathcal{T}_{\pi_k}^2(\boldsymbol{v}_k) \le \cdots \le \mathcal{T}_{\pi_k}^\infty(\boldsymbol{v}_k) = \boldsymbol{v}^{\pi_k} \le \boldsymbol{v}^\star.$$

This completes the inductive step.

Consequently, for $k = K = \lceil\log_2(\varepsilon^{-1}(1-\gamma)^{-1})\rceil$ iterations, $\varepsilon \ge \alpha_K \ge \varepsilon/4$ and with probability $1 - \delta$, $v_K$ is an $\varepsilon$-optimal value and $\pi_K$ is an $\varepsilon$-optimal policy.

For runtime and sample complexity, note that the algorithm can be implemented using only $\tilde{O}(N_K) = \tilde{O}((1-\gamma)^{-3}\varepsilon^{-2} + (1-\gamma)^3)$-samples and time per state-action pair. For the space complexity, note that the algorithm can be implemented to maintain only $O(1)$ vectors in $\mathbb{R}^{\mathcal{A}_{\text{tot}}}$. ∎

