# OpenReview forum: "Truncated Variance Reduced Value Iteration"
_NeurIPS.cc/2024/Conference — NeurIPS 2024 poster_

### Official Review · Reviewer_fJN4 · 2024-07-09

**Soundness:** 3
**Presentation:** 2
**Contribution:** 3
**Rating:** 5
**Confidence:** 3

**Summary:**

This paper proposes a new faster randomized algorithms for computing an $\varepsilon$-optimal policy in a  $\gamma$. discounted MDP The authors give an $\tilde{O}(A_{\text {tot }}\[(1-\gamma)^{-3} \varepsilon^{-2}+(1-\gamma)^{-2}])-$ time algorithm in the sampling setting, where the transition matrix is unknown but accessible through a generative model which can be called in $\tilde{O}(1)$ time, and  in the offline setting an $\tilde{O}\left(s+A_{\text {tot }}(1-\gamma)^{-2}\right)$-time algorithm where the probability transition matrix is known and $s$-sparse. This bound is attained using stochastic variance-reduce value iteration methods. Moreover, they provide a variant that carefully truncates the progress of its iterates to improve the variance of new variance-reduced sampling procedures. The advantage of their method is that in model-free that it can be implemented in $\tilde{O}\left(\mathcal{A}_{\text {tot }}\right)$ space when given generative model access.

**Strengths:**

1) The question of improving the computational cost per sample in computing optimal policies in DMDPs is a central question to tackle in MDPs.
4) The upper bound in terms of number of queries is state of the art for model free algoriths, for large $\epsilon$.
2) Maths are well written and proofs seem correct for me.
3) The truncated variance VI is a nice idea and may also lead to practical algoithms with lower computational cost.

**Weaknesses:**

4) The paper is sometimes difficult to follow.
5) I know that it is a theoretical work but a simple example on finite MDPs would be interesting to validate the computational cost and validate the new truncated variance VI algorithm.

**Questions:**

6) Do you  think it may be possible to bridge the sample complexity gag between model-based and model-free method using variance reduction techniques ?

**Limitations:**

No limitation.

---

> ### Author Rebuttal · Authors · 2024-08-07
>
> Thanks for your comments! We addressed your comment about experiments in the overall response and discuss individual questions here.
>
> **Comment 1: The paper is sometimes hard to follow.**
>
> We would be receptive to refining any areas that were challenging to follow. Could you please clarify where the writing made following challenging? We are happy to discuss more in the discussion period!
>
> **Q1: Do you think it may be possible to bridge the sample complexity gap between model-based and model-free methods using variance-reduction techniques?**
>
> This is a great question, and a key open problem! By narrowing the gap between model-based and model-free methods, we believe our paper provides hope that variance-reduction techniques or variants (such as our recursive-variance reduction plus truncation procedure) can be applied to eventually close the gap.
>
> Currently, the _only model-free_ methods that obtain optimal sample complexity for _some_ $\epsilon$ regime leverage variance reduction in some form or another, so it seems like it is a very powerful tool. Our work _reduces_ the sample complexity gap between model-based and model-free methods, and we _reduced_ this gap by _combining_ our new ideas (recursive variance reduction and truncation) with ideas from the previous state-of-the art [2, 3]. So, it is perhaps natural to hope that our techniques can be further combined with a few more tricks to close the gap entirely, but this is an open problem. This is an exciting direction for future work, and we hope our work provides useful tools for tackling this research question.

---

### Official Review · Reviewer_r2NT · 2024-07-09

**Soundness:** 3
**Presentation:** 2
**Contribution:** 3
**Rating:** 7
**Confidence:** 3

**Summary:**

The paper considers the problem of finding an $\varepsilon$-optimal policy of a discounted Markov decision process. Under the generative model, they propose a new algorithm with an improved the sample and time complexity. They also propose an extension to the case where the probability transition matrix is known and $s$-sparse. The core idea of the paper is to use a new variant of variance reduced learning called the truncated variance reduced learning, where the updates are truncated to ensure that the variance of the updates can be controlled.

**Strengths:**

I like the core idea of the paper to use truncation to control variances in the recursive variance reduction. Although simple, it seems to be quite effective.

**Weaknesses:**

Please see the next sections for my comments.


A minor comment: There seem to be a lot of typos throughout the paper, both in terms of formatting issues and spelling errors. I would suggest the authors to carefully go through the paper and address all of them for the final version.

**Questions:**

1.  Can the authors explain how they claim that model based methods use $\Omega(\mathcal{A}\_{\text{tot}}(1-\gamma)^{-3} \varepsilon^{-2})$ memory? The model based methods do not need to store all samples and can simply store the probability transition matrix which takes $\mathcal{O}(|\mathcal{S}| \mathcal{A}_{\text{tot}})$ memory.

2. The advantage of this work over that in [3] (as cited in paper) is mainly that truncation directly allows to control the burn in costs. By ensuring that one uses only $\mathcal{O}((1-\gamma)^{-2})$ samples in each iteration, one directly gets the leading term of $\mathcal{O}((1-\gamma)^{-3})$ and the burn-in of one iteration. On the other hand, in [3], the burn in is $\mathcal{O}((1-\gamma)^{-3})$ (corresponding to sample complexity of one iteration) and the leading term is controlled by careful analysis using variance of the optimal value function. Is my understanding correct? It seems that the analytical simplification is what is at the core of the contribution. Is there something else that is happening at a more fundamental level or just that the new approach offers improved analysis?

**Limitations:**

Yes

---

> ### Author Rebuttal · Authors · 2024-08-07
>
> Thank you for your feedack! We’ll carefully address the misspellings/typos ahead of the camera-ready.
>
> **Q1: Can the authors explain how they claim that model-based methods use $\Omega(A_{tot} (1-\gamma)^{-3} \epsilon^{-2})$?**
>
> Good point, indeed, the model-based methods actually only require $\Omega(A_{tot}  \min( (1-\gamma)^{-3} \epsilon^{-2}, |\mathcal{S}|))$-space. So, in the regime where $|\mathcal{S}| < (1-\gamma)^{-3} \epsilon^{-2}$, you are correct that the space of the model-free methods could be better. We will be certain to correct/clarify this nuance in the camera-ready, and thank you for pointing it out.
>
> **Q2: The advantage of this work over that in [3] as cited in the paper is that truncation directly allows us to control the burn-in costs by ensuring that one uses just ${O}((1-\gamma)^{-2})$ samples per state-action pair per iteration, whereas in [3], one required $O((1-\gamma)^{-3})$ per iteration and the leading term was controlled by careful analysis of the variance of the optimal value vector. Is my understanding correct, or is there something happening at a more fundamental level?**
>
> We addressed this at a high-level in the overall rebuttal response, but we’ll elaborate a bit here. Your perspective has a lot of alignment with our perspective, but we’ll just elaborate on a few points where our perspectives might differ.
>
> Recall, the model-free methods [ours and [3]] run $\tilde{O}(1)$ rounds, where each round halves the error in some initial value $v^{(0)}$. Each round works as follows (at a high level):
>
> * Estimate $Pv^{(0)}$ using samples
> * For each of $t = 1, 2, 3, ... , T = \tilde{\Theta}(1/(1-\gamma))$ iterations
>   * Maintain estimates of $P(v^{(t-1)} - v^{(0)})$
>   * Run one step of approximate variance-reduced value iteration using these estimates
>
> First, you correctly noted if the leading order term (number of samples required in Step 1.) is just controlled using the bound on the variance of the optimal values similar to what was done [3]. Indeed, this is exactly how we bound the leading order term.
>
> Second, you correctly noted that our main improvement is that we manage to maintain step (2a) using just $O((1-\gamma)^{-2})$ samples per state-action pair per iteration using a more sophisticated maintenance procedure. In contrast, [3] required $O((1-\gamma)^{-3})$ per iteration to maintain step (2a)). _However our improvement doesn’t come directly from truncation alone_. Concretely, our method differs from [3] in the following three ways:
> - _What [3] does_:
>   - In [3], Step (2b) is maintained by just estimating $P(v^{(t-1)} - v^{(0)})$ with new samples in each iteration.
> - _How we maintain Step (2b) with fewer samples:_
>   - **Recursive variance reduction:** At step $t$, notice that estimating the entire difference $P(v^{(t-1)} - v^{(0)})$ from scratch might be inefficient. We instead just estimate the marginal difference $P(v^{(t-1)} - v^{(t-2)})$. Then, we _leverage all our previous estimates from all the previous iterations_: $P(v^{(t-2)} - v^{(t-3)}), ...,   (v^{(1)} - v^{(0)})$. These automatically telescope to $P(v^{(t-1)} - v^{(0)})$.
>   - **Truncation:** Recursive variance reduction on it’s own doesn’t yield a theoretical improvement; but intuitively, it is useful if we can make sure that the entrywise maximum of $P(v^{(t)} - v^{(t-1)})$ is quantitatively smaller _entrywise_ than $P(v^{(t-1)} - v^{(0)})$. We do this by using truncation to ensure that our worst-case bound on $P(v^{(t-1)} - v^{(t-2)})$ is a $(1-\gamma)$ factor smaller than our worst-case bound on $P(v^{(t-1)} - v^{(0)})$.
>   - **Analysis:** To show that the preceding algorithmic ideas provably lead to an improved sample complexity, we need to model the recursive variance reduction procedure as a martingale and use Freedman’s inequality.
>
> So, in summary, two algorithmic changes and one analytic change enable our improvement-- truncation is just one piece.

---

> > ### Comment · Reviewer_r2NT · 2024-08-09
> > **Response to the authors**
> >
> > Thank you for the detailed response. That clarifies all my concerns. I will maintain my score.

---

### Official Review · Reviewer_FfgG · 2024-07-12

**Soundness:** 4
**Presentation:** 3
**Contribution:** 4
**Rating:** 8
**Confidence:** 4

**Summary:**

The paper provides a randomized algorithm to compute nearly-optimal policies in DMDPs (in the bounded-reward setting) for regimes where the probability transition matrix is either known or unknown. The sample complexities in the paper removes a multiplicative factor of $\frac{1}{1-\gamma}$ on one of the terms in each regime. The algorithm provided is model-free and only requires $\tilde{O}(|A|)$ space. The main claims are all rigorously supported by proofs to verify this work, and (broadly speaking) this work is an important stepping stone towards closing the sample-complexity gap between model-free and model-based methods.

**Strengths:**

1) In the sample setting, the algorithm runs in nearly linear time in the number of samples, for a particular choice of $\epsilon$, which is a novel result.
2) The analysis differs from existing works in the way the utilities are computed - this work uses a recursive variance reduction method.
3) In general, the proofs are non-trivial and quite hard to follow, emulating much of a TCS style of writing - but they are all quite rigorous and well-resented. The method uses an elegant construction of $\nu^{(t)}$ as a supermartingale and adapts Freedman’s inequality with some of the methods from the large-deviations literature. It is interesting that this rather simplistic change in approximating the utilities $g^{(t)}$ results in the drop of the $\frac{1}{1-\gamma}$ multiplicative term!

**Weaknesses:**

I wonder if the constants in Lemma 2.2 and in the burn-in phase in algorithm 6 can be strengthened? A constant of $10^4$ in the term $N_{k-1}$ might not be so practical... I will emphasize that this is of less concern since this paper primarily focuses on theoretical results and mathematical analyses. Beyond this, I do not see any other potential weaknesses. I checked the math as well and it seems sound.

**Questions:**

See weaknesses.

**Limitations:**

None.

---

> ### Author Rebuttal · Authors · 2024-08-07
>
> Thank you for your feedback!
>
> **Comment/Question: Can the constant of $2^8$ in Lemma 2.2 or the constant of $10^4$ be tightened?**
>
> Regarding the comment about the tightness of the $10^4$ constant in $N_{k-1}$, from a quick calculation, we believe it can be lowered to roughly 6500. We think further improvements in this leading constant should also be possible, but this may come at the cost of slightly larger log factors (e.g., from the analysis in the proof of Thm. 1.1.) because in the proof we have some flexibility to trade off whether constants appear up-front or inside of polylog factors in $\epsilon, \delta, |\mathcal{S}|, \mathcal{A}_{tot}, (1-\gamma)$.
>
> Some improvement might be possible in the case of the constant of $2^8 = 256$ in Lemma 2.2, but it is less clear how to do this without paying larger constants inside polylog factors in $\epsilon, \delta, |\mathcal{S}|, \mathcal{A}_{tot}, (1-\gamma)$.
>
> As you already noted, we did not try to reduce constants too much as it wasn’t the focus of our work; however, we will certainly make an effort to tighten the constants where possible ahead of the camera-ready! Thanks for the suggestion!

---

> > ### Comment · Reviewer_FfgG · 2024-08-11
> > **Regarding constants**
> >
> > Thanks for the reply!
> > It would be great if the authors could include a discussion about the constants in the camera-ready version. This tradeoff between constant factors and polylog factors in $\epsilon,\delta$, etc is certainly interesting...

---

> ### Author Response · Authors · 2024-08-14
> **Thanks for your response!**
>
> Thanks for your reply! We will be sure to tighten constants where possible, and we will also be sure to include a discussion of the constants in the camera-ready version of our paper. Thank you for the suggestion!

---

### Official Review · Reviewer_ENzu · 2024-07-13

**Soundness:** 3
**Presentation:** 3
**Contribution:** 3
**Rating:** 7
**Confidence:** 3

**Summary:**

This paper introduces Truncated Variance-Reduced Value Iteration (TVRVI), which enhances the previous prior state-of-the-art sample complexity for computing an $\epsilon$-optimal policy in both offline and sampling settings. Specifically, in the offline setting, where the probability transition matrix is known, Theorem $1.2$ established a sample complexity of $\tilde{O}(nnz(P)+\mathcal{A}(1-\gamma)^{-2})$ which improves previous bound of $\tilde{O}(nnz(P)+\mathcal{A}(1-\gamma)^{-3})$ sample complexity. In a sampling setting, where the probability transition matrix is unknown but accessible through a generative model, Theorem $1.1$ established a sample complextiy of $\tilde{O}(\mathcal{A}((1-\gamma)^{-3}\epsilon^{-2}+(1-\gamma)^{-2})$, improving upon previous  $\tilde{O}(\mathcal{A}((1 \gamma)^{-3}\epsilon^{-2}+(1-\gamma)^{-3})$ bound. In both settings, TVRVI requires $\tilde{O}(\mathcal{A})$-space, provided suitable access to the input, and aim to efficiently compute a coarse approximation of the optimal policy for large $\epsilon$.

**Strengths:**

Incorporating Freedman’s analysis, truncation, and other variance reduction techniques from prior works, this work provides a state-of-the-art bound on sample complexity. The high-level idea is demonstrated, and detailed comparisons with prior work are provided. It's worth mentioning that in the literature on the theoretical analysis of sample complexity, the application of increasingly generalized inequalities —progressing from Hoeffding [2] to Bernstein [3], and now to the Freedman inequality in this work— has led to incremental improvements in theoretical results.

**Weaknesses:**

There are no experimental results, and I am uncertain about the ignorance of the polylogarithmic factors involving  $\epsilon$ and $\delta$. Also, in the comparison tables, the range of $\epsilon$ is not the best.

**Questions:**

(0) Are these theoretical results the state-of-art even if $\epsilon$ is small?

(1) Could you clarify why it's okay to ignore polylogarithmic factors involving  $\epsilon$ and $\delta$?

(2) I wonder if this kind of theoretical sample complexity results lead to practical benefits or is more meaningful from a theoretical perspective. For instance, in convex optimization, Nesterov's accelerated gradient descent demonstrates theoretical acceleration and holds significance in the literature of complexity lower bound rather than for its practical applications (While adaptive Nesterov's type algorithm like Adam works well in deep learning, I believe Nesterov's acceleration is more meaningful from a theoretical perspective),

 (3)  In line $5$ of Algorithm $1$, could you explain the intuition of the offset parameter and subtraction with different powers?

(4) Could these analyses be extended to the settings where the reward is unknown or average reward MDP?

 (5) In line 27, why do we need condition $|\mathcal{A}| \ge |\mathcal{S}| $?


 (6) In line 98, does `nearly-linear' indicate quadratic?

 (7) In line 100, why it is $\epsilon$? If $\epsilon = O(1-\gamma)^{-1/2}$, isn't it $\epsilon^4$?

(8) References [1] and [2] seem the same.

(9) References [17] and [21] seem the same.

(10) In line 104, what is the definition of $w$?

(11) In line 191, $v^{(t-1)}$ should be changed to $v^{(t)}$

(12) In line 193, typo $v^{(0)}$

(13) In line 276, typo $v^{(0)}$




 (14) In line 282, need spacing.

**Limitations:**

Yes, the authors addressed limitations.

---

> ### Author Rebuttal · Authors · 2024-08-07
>
> Thanks for the feedback! We responded about experiments above and discuss your questions here.
>
> **Applying stronger inequalities-- Hoeffding [2], Bernstein [3], Freedman [this paper] has led to improvements.**
>
> As discussed in the main rebuttal, Freedman _alone_ is insufficient to obtain our improvements. We also needed two _novel algorithmic changes_ (truncation + recursive variance reduction.) Our key insight is the _combination_ of stronger algorithmic tools plus improved concentration analysis.
>
> **I’m uncertain about ignoring polylogarithmic factors.**
>
> Please see Q1 below.
>
> **The range of $\epsilon$ is not the best in the tables.**
>
> In _Table 1_, all methods have polylog $\epsilon$ dependence, so there’s no $\epsilon$ regime constraint. In _Table 2_, [18] has a larger $\epsilon$ range. However, Table 2 shows just an improved $\epsilon$ range for _query complexity, not runtime_. As discussed in Lines 48-77, [18] is _model-based_, so it has higher time and space complexity than ours.
>
> **Q0: Are the theoretical results state-of-the-art even for small $\epsilon$?**
>
> Yes, we _match_ state-of-the-art optimal algorithms in the small $\epsilon$ regime (up to logs) and _improve_ for large $\epsilon$ regime [Lines 91-94].
>
> **Q1: Why can we ignore polylogarithmic factors in $\epsilon, \delta$?**
>
> It’s standard to hide polylogarithmic factors (polylogs) in the parameters $\epsilon$, $\delta, (1-\gamma)$, $|\mathcal{S}|$, and $\mathcal{A}_{tot}$ inside tilde-notation $(\tilde{O}/\tilde{\Omega})$ in this line of research (as in much machine learning, optimization, and algorithm design theory literature). The long line of work on this problem [2, 3, 17, 18, 24] follow this same convention when comparing methods; thus, ignoring polylogs is natural/important for comparing prior state-of-the-art results. Note the polylogs hidden inside tilde-notation _are not hiding exponential terms or new problem parameters_. Moreover, these polylogs are quantified explicitly in the main body (in all “algorithm” environments and intermediate lemmas) for full transparency. We may investigate ways to make these polylogs more explicit in the formal statements of Thm. 1.1 and Thm. 1.2 in the camera-ready version. To briefly explain why this is convention: polylogs scale much better than polynomial factors, so, when designing algorithms with theoretical guarantees, we focus on improving polynomial factors and not the, relatively, lower order polylogs.
>
> **Q2: Do the methods lead to practical benefits aside from theoretical benefit (e.g., AGD vs. ADAM)?**
>
> Our motivation is the theoretical perspective (as in [3, NeurIPS ‘18] and [17, NeurIPS ‘20] and [15, NeurIPS ‘98]) of better characterizing the fundamental mathematical limits of what runtime/space/sample complexities can be achieved for RL with a generative model. It’s possible our techniques may motivate further practical improvements/analogs, but this is outside the scope of our current work.
>
> **Q3: Why the offset parameter in Alg. 2?**
>
> Great question! We didn't have space in the submission, but plan to use the extra page of camera-ready to include a discussion of this.
> - Why the offset? The offset shifts estimates of the expected utilities _down_ so that the _shifted estimates_ are _underestimates_ of the true expected utilities. This technique is adapted from [3] and lets us convert optimality guarantees on values to optimality guarantees on policies. If our estimated expected utilities are always underestimates, then one can show that at each iteration _our current value vector estimate is an _underestimate_ of the true value of the current policy._ Note that this ensures that if our current _value_ is $\epsilon$-optimal, the policy must _also_ be _at least_ $\epsilon$ optimal! As a further note, this offset enables us to get fine-grained bounds in Section 3 and Appendix A (this arises in the proofs of Theorem 1.1 and Theorem A.1).
> - Why those specific powers? If we use Berstein to guarantee that we _shifted our original estimate down_ far enough be an _underestimate_ of the true utilities, then these terms are what we need to subtract in order to succeed with probability $1-\delta$ (see for instance, in the proof of Lemma 3.1.)
>
> **Q4: Do the methods extend to average-reward or unknown rewards?**
>
> Great question! [A] reduced solving average-reward MDPs (AMDPs) to solving discounted MDPs for a sufficiently high discount factor, so our method can apply to AMDPs. We also believe the techniques might be amenable to unknown rewards, but this is an open question. We believe the previous works cited in our tables do not directly consider the unknown reward case either, so it is slightly outside the scope of our work. [A] Yujia Jin and Aaron Sidford. Towards Tight Bounds on the Sample Complexity of Average-reward MDPs. ICML ‘21.
>
> **Q5: Why should $|\mathcal{A}| \geq \mathcal{S}|$?**
>
> $\mathcal{A}$ is _all state-action pairs_ so $|\mathcal{A}| \geq |\mathcal{S}|$ just ensures we have at least one action available in each state.
>
> **Q6: What is nearly-linear in Line 98?**
>
> Here, it means $\tilde{O}(A_{tot}(1-\gamma)^{-2}\epsilon^{-3})$ because it is equal (up to logs and constants) to the optimal number of samples, $\tilde{\Omega}(A_{tot}(1-\gamma)^{-2}\epsilon^{-3})$.
>
> **Q7: Can you explain Line 100?**
>
> Thm. 1.1 runs in $\tilde{O}(A_{tot}[(1-\gamma)^{-2}\epsilon^{-3} + (1-\gamma)^{-2}])$. Suppose $\epsilon = O((1-\gamma)^{-1/2})$. Then, the $\tilde{O}(1-\gamma)^{-2}$ is lower order, and the runtime is $\tilde{O}(A_{tot}(1-\gamma)^{-2}\epsilon^{-3})$, which matches the lower bound of $\tilde{\Omega}(A_{tot}(1-\gamma)^{-2}\epsilon^{-3})$ up to logs/constants.
>
> **Q10: What is w in Line 104?**
>
> It is little-omega. A function $f(n)$ is $ \omega(1)$ if $\lim_{n \rightarrow \infty} f(n) = \infty$. We’ll add a definition in camera-ready!
>
> **Q8-9, Q11-14: Typos**
>
> Thanks for pointing out the duplicate references and typos. We’ll correct in the camera ready!

---

> > ### Comment · Reviewer_ENzu · 2024-08-14
> > **Response to the authors**
> >
> > Thank you for addressing my questions in your rebuttal. I will maintain my score.

---

### Author Rebuttal · Authors · 2024-08-07

We thank the reviewers for their thoughtful feedback and questions! We are encouraged that the reviewers had an overall positive view of our work. In particular, we appreciate that reviewers found our truncated variance reduction idea to be nice as well as novel and felt that the problem we study is central/important.

We address two of the main comments in the reviews below and respond to reviewer-specific clarification in the reviewer-specific rebuttal responses.

1. **Two Reviewers asked if the analytical improvement (due to our improved martingale analysis) was sufficient to obtain our improvement and expressed curiosity about whether truncation or martingale analysis alone is sufficient.**

Thank you for the question! To recap, as discussed in the approach section of our paper [Lines 187-232], we combined two novel algorithmic ideas (1) truncation and (2) recursive variance reduction to obtain our improvements. To analyze these two algorithmic ideas and prove that they lead to improvement, we use (3) a new analytical approach: martingale analysis via Freedman’s inequality.

We believe all three ingredients are necessary for the analysis to work. We obtained our result only by the _combination of both of these two new algorithmic ideas along with the new analysis idea_. Thus, our novelty is not only the new martingale analysis but also the new algorithmic ideas. Moreover, to the best of our knowledge, using just one of the algorithmic ideas is insufficient. That is, to the best of our knowledge,
- Using just truncation without recursive variance reduction is not strong enough for the proofs to go through.
- Using just recursive variance reduction without truncation is also not strong enough.

2. **Two reviewers noted that our work does not include experiments.**

We agree that the results of an extensive empirical comparison of various model-free and model-based algorithms for this problem would be an intriguing research direction. One potential obstacle in this direction is that, because the related/cited work is also purely theoretical, we are unaware of the extent to which previous state-of-the-art algorithms for the problem have been implemented (e.g., [18 NeurIPS 2020], [17 COLT 2020], and [24 Mathematics of Operations Research 2019]). While we share the reviewers’ enthusiasm for seeing a comprehensive comparison of the variety of state-of-the-art algorithms on synthetic and real-world MDPs; this is outside the scope of our submission.

We would be happy to answer further questions during the discussion period. Thanks again to the reviewers!

---

### Decision · Program_Chairs · 2024-09-25

**Decision:**

Accept (poster)

**Comment:**

The paper improves upon the existing results on variance reduced model free methods for RL by introducing two algorithm techniques of truncation and recursive variance reduction and the using Freedman's inequality for concentration of martingales. This brings the sample complexity of model free methods (which are significantly more efficient in runtime and memory requirements) to methods that sample and build a model. These improvements apply in the range of epsilon >> 1. Overall the reviewers unanimously appreciated the contributions of the paper as well as the writing and exposition. The improvement shown in the paper are clear, concrete and interesting to a broad RL theory community. The techniques introduced are also interesting to the wider community in my opinion. I am happy to recommend acceptance. I ask the authors to incorporate the comments and use the extra page for more explanations of the pieces as promised.